



# Pulsed carbon export from mountains by earthquake-triggered landslides explored in a reduced-complexity model

Thomas Croissant[1], Robert G. Hilton[1], Gen Li[2], Jamie Howarth[3], Jin Wang[1,a], Erin L. Harvey[1,b], Philippe Steer[4] and Alexander L. Densmore[1]

[1]Department of Geography, Durham University, Durham, DH1 3LE, United Kingdom
[2]Division of Geological and Planetary Sciences, California Institute of Technology, Pasadena, CA, 91125, USA
[3]School of Geography, Environment and Earth Sciences, Victoria University of Wellington, PO Box 600, Wellington, New Zealand
[4]Universite de Rennes 1, CNRS, Géosciences Rennes – UMR 6118, 35000 Rennes, France
[a]Now at: SKLLQG, Institute of Earth Environment, Chinese Academy of Sciences, Xi'an 710061, China.
[b]Now at: School of Earth and Ocean Sciences, Cardiff University, Main Building, Park Place, Cardiff, CF10 3AT, UK.

*Correspondence to*: Thomas Croissant (thomas.croissant@durham.ac.uk)

**Abstract.** In mountain ranges, earthquakes can trigger widespread landsliding and mobilise large amounts of organic carbon by eroding soil and vegetation from hillslopes. Following a major earthquake, the landslide-mobilised organic carbon can be exported from river catchments by physical sediment transport processes, or stored within the landscape where it may be degraded by heterotrophic respiration. The competition between these physical and biogeochemical processes governs a net transfer of carbon between the atmosphere and sedimentary organic matter, yet their relative importance following a large landslide-triggering earthquake remains poorly constrained. Here, we propose a model framework to quantify the post-seismic redistribution of soil-derived organic carbon. The approach combines predictions based on empirical observations of co-seismic sediment mobilisation, with a description of the physical and biogeochemical processes involved after the earthquake. Earthquake-triggered landslide populations are generated by randomly sampling a landslide area distribution, a proportion of which is initially connected to the fluvial network. Initially disconnected landslide deposits are transported downslope and connected to rivers at a constant velocity in the post-seismic period. Disconnected landslide deposits lose organic carbon by heterotrophic oxidation, while connected deposits lose organic carbon synchronously by both oxidation and river export. The modelling approach is numerically efficient and allows us to explore a large range of parameter values that exert a control on the fate of organic carbon in the upland erosional system. We explore the role of the climatic context (in terms of mean annual runoff and runoff variability) and rates of organic matter degradation using single and multi-pool models. Our results highlight that the redistribution of organic carbon is strongly controlled by the annual runoff and the extent of landslide connection, but less so by the choice of organic matter degradation model. In the context of mountain ranges typical of the southwest Pacific region, we find that model configurations allow for more than 90% of the landslide-mobilized carbon to be exported from mountain catchments. A simulation of earthquake cycles suggests efficient transfer of organic carbon out of a mountain range during the first decade of the post-seismic period. Pulsed erosion of organic matter



by earthquake-triggered landslides therefore offers an effective process to promote carbon sequestration in sedimentary deposits over thousands of years.

## 1 Introduction

Erosion of carbon from the terrestrial biosphere results in an important transfer of carbon from land to oceans by rivers (Galy et al., 2015; Stallard, 1998). Some of the eroded organic matter escapes degradation and can be buried in sediments (Burdige, 2005; Galy et al., 2007), contributing to a long-term sequestration of atmospheric carbon dioxide ($CO_2$) (Hilton & West, 2020). To constrain how the export of organic carbon (OC) from land may vary over space and time and force the carbon cycle, we need to understand the factors that drive its erosion from the landscape and transfer by rivers (Berhe et al., 2018; Hilton, 2017). In mountain ranges located at convergent plate boundaries, large earthquakes ($M_w > 6$) can trigger widespread landsliding (Keefer, 1984; Tanyaş et al., 2017) which harvest OC by mobilising soil and vegetation from hillslopes (Allen et al., 1999; Garwood et al., 1979). Earthquakes and other events which trigger widespread disturbance of forests, such as storms, have been considered as a $CO_2$ source, due to degradation of the eroded organic matter by heterotrophic respiration (Chambers et al., 2007; Zeng et al., 2009). However, recent work has highlighted that landslide-mobilised OC can be exported from mountain catchments by rivers (Hilton, et al., 2008; Wang et al., 2016) and that this export can be sustained over thousands of years, as recorded in lake deposits (Howarth et al., 2012; Frith et al., 2018). When coupled to the re-growth of the forest and soil carbon stocks at the site of erosion, this drives net $CO_2$ drawdown from the atmosphere (Berhe et al., 2007; Stallard, 1998). Despite this recognition, it remains a challenge to constrain how variability in climate (e.g., annual runoff) and biogeochemical processes (such as respiration) control the fate of landslide-mobilized OC, and the timescales over which their impacts are felt (Ramos Scharrón et al., 2012; Restrepo et al., 2009). We require better constraint on the role of climatic and biogeochemical processes over timeframes that span multiple seismic cycles. To do this, we require a theoretical framework to explore the competition between physical and biogeochemical processes from the site of erosion to the catchment outlet.

The export of OC from mountain river catchments is linked to that of fine clastic sediment (Galy et al., 2015; Hilton et al., 2012). Following widespread earthquake-triggered landsliding, fine sediment can be evacuated as fluvial suspended load within a decade in wet, subtropical settings such as Taiwan (Dadson et al., 2004; Hovius et al., 2011), or over several decades in more arid locations (Tolorza et al., 2019; Wang et al., 2015). The total landslide-derived sediment volume, including coarse material which can be mobilized by debris flows and transported as bed load, may take decades to centuries to export (Croissant et al., 2017; Croissant et al., 2019; Fan et al., 2018; Yanites et al., 2010). This variability in the temporal dynamics of post-seismic sediment transfer highlights the role of factors such as the connectivity of sediment to the fluvial network, and the rate of transport during the post-seismic phase (Wang et al., 2015). The timescales of sediment routing are key to understanding the fate of the landslide-eroded OC. Any delay in transport provides time for OC to be degraded in the



landscape (Berhe et al., 2007; Ramos Scharrón et al., 2012). This could happen, for example, if OC is exposed on the surface of landslide deposits or colluvium, or during transient storage of eroded materials in fluvial bars.

In this study, we introduce a reduced-complexity modelling approach to quantify the fate of OC eroded from soils after a widespread landslide-triggering event, such as a major earthquake. Our aim is to explore the climatic, tectonic and

biogeochemical controls on the fate of OC over decadal to centennial timescales. To do this, we describe the evolution of OC in landslide deposits by two principal processes: i) the physical transport of OC as suspended load by fluvial export, which can act as a potential $CO_2$ sink; and ii) the oxidation of OC by heterotrophic respiration  and subsequent $CO_2$ release. We explore several approaches to quantify organic matter degradation using different biochemical models described in the literature. We conceptualize our model based on the observed processes and boundary conditions operating in the western

Southern Alps of New Zealand. A key concept is that of landslide connection to the fluvial network, and the potential evolution of that connection through time following a large earthquake. We consider time scales from the occurrence of a single landslide to several earthquake cycles, and explore the climatic boundary conditions (mean annual runoff and runoff variability) which impact the fate of landslide-mobilized OC.

## 2 Methods

We use a reduced-complexity model approach to quantify the fate of the landslide-mobilized OC, considering the case of earthquake-triggered landslides. The model Quakos is described at length by Croissant et al. [2019]. Herein, we detail the components involved in this study, and present the biochemical models used to explore the fate of eroded OC. The model considers the erosion and transport of fine grained, soil-derived OC, with a diameter that is nominally ~ 1 mm or finer. This material contributes mostly to the suspended sediment load of mountain rivers (Clark et al., 2017; Hatten et al., 2012; Hilton,

Galy, & Hovius, 2008) and can be exported to large river systems (Bouchez et al., 2014) and sedimentary deposits (Galy et al., 2007). At present we cannot simulate the movement of coarse OC supplied by erosion of living biomass (and any associated coarse woody debris), mainly due to the lack of robust transport laws (Wohl, 2011). Recent work suggests that this coarser woody material could be a large component of erosional OC budgets (Mohr et al., 2017; Turowski et al., 2016; A. J. West et al., 2011). Indeed, it may be transported long distances and preserved in marine sediments (Lee et al., 2019).

We also do not consider rock-derived OC in the model framework. While the oxidation of rock OC may be substantial in the landscape (Hemingway et al., 2018; Horan et al., 2017), the degradation rates of this material remains poorly constrained (Chang & Berner, 1999) and so we do not attempt to model its contribution to the net $CO_2$ budget of erosion (Hilton & West, 2020). As such, the model results described herein do not capture the full impact of erosion on the fate of OC, but provide a framework that could be adapted in future to include these additional carbon transfers.



## 2.1 Study area

A conceptual framework is provided by a hypothetical earthquake on the Alpine Fault in the western Southern Alps, New Zealand (Fig. 1). The topography of the west coast of the Southern Alps is obtained from the SRTM3 digital elevation model (DEM). This location is selected due to prior work documenting the role of landsliding for sediment and OC mobilisation (Hilton et al 2008; Hovius et al., 1997; Korup et al., 2010) and the role of large earthquakes in long-term sediment and OC records in lakes (Frith et al., 2018; Howarth et al., 2012, 2014). The Southern Alps formed as a result of oblique plate convergence between the Pacific and Australian plates at a rate of 39.7 mm.yr$^{-1}$ (DeMets et al., 2010). The Alpine fault accommodates up to 80% of the plate convergence (Norris & Cooper, 2007) in earthquakes of magnitude > 7.5, with the last major earthquake occurring in AD1717 and an average recurrence time of 263 ± 68 years as determined from paleoseismic studies (Howarth et al., 2016, 2018). The Southern Alps form a barrier to westerly winds that leads to high precipitation rates of up to 13 m yr$^{-1}$ along the west coast (Tait & Zheng, 2007). Landscapes in the Southern Alps are characterized by steep hillslopes with modal gradients of c. 35° (Korup et al., 2010) and hillslope erosion is dominated by landsliding in the current aseismic period (Hovius et al., 1997).

The high rainfall sustains temperate rainforest on steep slopes at elevations ≤ 800 m, while shrubs, herbs and grassland persist up to and above the regional snowline at ~1,250 m (Reif & Allen, 1988). The carbon stocks of above-ground biomass in the western Southern Alps have been estimated at 17,500 ± 5,500 tC.km$^{-2}$ based on local forest plot inventories (Hilton et al., 2011). The OC stocks of soils in the Southern Alps have been recently estimated at ~13,000 ± 4300 tC.km$^{-2}$ (Harvey, 2019). This estimate was derived from measured OC concentrations in soil profiles collected from 52 sites in the mountain range. The OC stocks are estimated to the point of refusal of the soil auger profiles (average thickness = 0.42 ± 0.09 m) and comprise soil OC in the upper organic-rich horizons (mean thickness = 0.10 ± 0.01 m) and in the mineral-dominated B and C horizons beneath. The average value is slightly lower than previous estimates which were inferred from sparse data (Frith et al., 2018; Hilton et al., 2011; Tonkin & Basher, 2001).

## 2.2 Generation of earthquake-triggered landslide clusters

Populations of landslides triggered by large earthquakes are generated in the Quakos model, with an empirical relationship between the spatial patterns of the simulated peak ground acceleration (PGA) and landslide density (see Croissant et al., 2019 for further details). The resulting landslide density map allows for the quantification of the total area of landsliding ($A_{ls,tot}$) and determines the spatial extent of mass wasting. The co-seismic landslides are described by several components: their geometrical properties of their scar area ($A_{ls}$, m$^2$) and deposit volume ($V_{ls}$, m$^3$), their spatial distribution within the landscape, and their connectivity to the drainage network. Based on previous empirical studies (Malamud et al., 2004; Tanyaş et al., 2017), the full distribution of landslide areas can be described by a three-parameter inverse gamma function of the form:



$$pdf\left(A_{ls}\right)=\frac{1}{a\Gamma\left(a\right)}\left[\frac{a}{A_{ls}-s}\right]^{\rho+1}\exp\left[\frac{-a}{A_{ls}-s}\right],$$ (1)

where $a$ is the position of the pdf maximum, $s$ controls the roll-over for small landslides, $\rho$ is a positive exponent controlling the slope of the tail and $\Gamma$ is the Gamma function. The parameterization of equation 1 has been done using values obtained for a West Coast landslide inventory (Hovius et al, 1997). In Quakos, a landslide population is generated by sampling the pdf until the sum of the landslide areas reaches the value of $A_{ls,tot}$. The landslides are spatially distributed within catchments following a landslide density map that depends on the PGA patterns triggered by the earthquake (Meunier et al., 2008). Landslide area is converted to volume with an empirical law of the form:

$$V_{ls}=\alpha\,A_{ls}^{\gamma},$$ (2)

where the empirical constants $\alpha$ and $\gamma$ are set to 0.05 and 1.5 (Hovius et al., 1997; Larsen et al., 2010) to reflect the deep-seated bedrock landslides that are likely to be the dominant mechanism of landslide generation during large earthquakes in New Zealand (Larsen et al., 2010). $V_{ls}$ is a measure of the total volume of sediment mobilized by landslides. However, in the subsequent modelling we consider only the fine-grained portion of this landslide volume that can be transported as suspended load ($V_{ls,fine}$), here set to 50% of $V_{ls}$, as an average value considering empirical estimations varying between 10 and 90% (Dadson et al, 2003). The landslide volume is distributed in a single model cell.

Each landslide is assigned a probability that it is initially connected to the fluvial network ($C$), which is a function of the drainage area of landslide pixels following an empirical relationship of the form (e.g. 2008, Wenchuan (Li et al., 2016) and 2015, Gorkha (Roback et al., 2018)):

$$C=c\,A_{ls}^{\mu},$$ (3)

where $c$ and $\mu$ are empirical constants. Equation 3 is used to assign the initial connectivity status of all landslides with an area smaller than $10^6$ m², while larger landslides are assumed to be initially connected (Croissant et al., 2019). Using $c$ = 0.86 and $\mu$ = 0.345, only 12% of the landslides are initially connected to the drainage network (Fig. 1). However, because the largest landslides are more likely to be initially connected, between 40 and 60% of the landslide-derived sediment volume is initially available. We assume that sediment derived from initially connected landslides is immediately available for fluvial export toward the catchment outlet.

To account for processes which mobilise sediment from landslide deposits that are not initially connected to the river network (including soil creep, overland flow, shallow landsliding, and debris flows), which are known to act (Dietrich et al., 2003; Roering et al., 2001) but are otherwise difficult to parameterise, our approach allows landslide deposits on hillslopes to progressively reach the river network at a rate defined as a constant "connection velocity" $u_{con}$ (m yr⁻¹) (Croissant et al.,



2019). The path taken by the landslides to reach the closest river is computed using a steepest descent algorithm and their time of connection ($t_{con}$) is determined by dividing the distance to the river network by $u_{con}$. The river network is extracted from the DEM using a single flow algorithm and by setting a critical drainage area of 0.5 km².

To quantify the erosion of soil OC, the area of each individual landslide scars is combined with the local soil organic carbon stock ($C_{stock}$, tC m⁻²) determined in this study to quantify the total mass of eroded soil OC (tC):

$$M_{oc,tot} = \sum_{0}^{N} A_{ls,n} C_{stock,n} \, , \tag{4}$$

where N is the total number of earthquake-triggered landslides. Simulations account for the uncertainties related to the OC content of soils. As outlined at the opening of Section 2, the transport of the above-ground biomass is not considered here as we assume it is mobilized as large woody debris, whose transport dynamics remain poorly constrained (Wohl, 2011).

## 2.3 Along-stream river transport capacity

The model framework considers that once a landslide deposit reaches the fluvial network, sediment and OC are transported at a rate set by the local transport capacity of the river. Here, we only consider the evacuation of the fine sediment and OC that are transported as suspended load, with a nominal grain size < 1 mm. First, we consider the event-based sediment discharge of the suspended load ($Q_s$, m³ s⁻¹) that is commonly empirically described as a power-law function of water discharge ($Q$, m³ s⁻¹):

$$Q_s = k_s Q^\varepsilon \, , \tag{5}$$

where $k_s$ is a constant (s m⁻³) and $\varepsilon$ is a positive exponent. Here, the water discharge is a linear function of the local drainage area and surface runoff. The constants $k_s$ and $\varepsilon$ can be determined from measurements across a range of discharges. For our study location, we use gauging data from the Hokitika, Haast, Whataroa and Poerua rivers, and find values of $k_s$ = 300 and $\varepsilon$ = 2. To encompass the contribution of all discharge events to sediment transport, the annual long term sediment flux is computed as the integral of the convolution between sediment flux (equation 5) and the probability density function of daily discharge events ($pdf(Q)$) (DiBiase & Whipple, 2011; Lague, 2014; Lague et al., 2005):

$$\overline{Q_s} = \int_{0}^{Q_{max}} Q_s \, pdf(Q) \, , \tag{6}$$

where $Q_{max}$ is the maximum discharge in the range. The pdf of daily discharges is described by an inverse gamma function of the form:





$$pdf\left(Q\right)=\frac{k^{k+1}}{\Gamma\left(k+1\right)}\exp\left(\frac{-k}{Q/\overline{Q}}\right)\left(Q/\overline{Q}\right)^{-(2+k)},\tag{7}$$

Where $k$ is a positive constant. We assume that the fine particulate OC has a transport behaviour which is similar to that of clastic suspended sediments, as demonstrated in a number of studies of turbulent mountain rivers (Clark et al., 2017; Hatten et al., 2012; Hilton, 2017; Smith et al., 2013). Thus, the transport of OC and fine sediment are proportional in the model, and

the rate of change of the mass of OC in a given landslide can be described by:

$$\frac{d\,M_{oc}}{dt}=-\overline{Q}_s\frac{M_{oc}}{M_{ls}},\tag{8}$$

where $M_{oc}$ is the mass of organic carbon mobilized by a landslide, $M_{ls}$ is the landslide sediment mass (assuming a sediment density of 2650 kg m⁻³) and $t$ is time. Finally, we introduce the characteristic time $t_0=\dfrac{V_{ls,fine}}{\overline{Q}_s}$ corresponding to the time necessary to remove all of the fine sediment and fine particulate OC mobilized by landslides.

**2.4 Sediment transport model assumptions**

We use a reduced-complexity approach to describe the physical processes in play, which allows us to explore the large spatial and temporal scales over which earthquake-triggered landslides impact landscapes. However, this means that the transport of fine sediment and particulate OC is subjected to several assumptions. First, by considering only the transport of fine, soil-derived particles as suspended load, each landslide volume is partitioned between fine grains (that can be

transported in river suspension) and coarse grains (which are transported by saltation and other bed load processes). Second, sediment transport is treated as a detachment-limited regime, and entrained particles are never re-deposited within the catchment. This assumption is likely to be reasonable for the Southern Alps, which are characterized by steep and short rivers that promote efficient suspended sediment transport (Korup et al., 2010). We also assume that the fine sediment is exported at fluvial transport capacity (equation 6). We do not consider in-situ stabilization of landslide materials, nor

regrowth of vegetation on those materials. Therefore, all landslide-derived fine grains will be eventually exported by rivers at the timescale controlled by the local transport capacity. In addition, the climatic context (mean annual runoff and runoff variability) that control the fluvial transport capacity are assumed to be constant over the duration of the simulations. Finally, any other event that might be triggering landslides during the post-seismic phase (rainstorms, earthquakes of lower magnitude, etc.,) are not considered.





## 2.5 Evolution of OC after landslide mobilization

Our model postulates that the fate of landslide-mobilized OC is controlled by its physical export by rivers versus its biogeochemical oxidation that can release $CO_2$ back to the atmosphere. The long-term sink of $CO_2$ by erosion comes about because a fraction of the mobilized OC is stored in sediments, while the vegetation and soil OC are renewed at the site of erosion (Berhe et al., 2007; Stallard, 1998). If eroded materials remain on hillslopes, the OC contained in the landslide deposits can be subjected to heterotrophic respiration and OC oxidation, either by soil fauna present in the original forest soil and translocated to the landslide deposit, or by colonisation by new fauna communities. When landslides are connected to the river network, the model framework considers that the OC remaining in the deposit experiences oxidation and physical transport simultaneously. OC oxidation can be generally described by either discrete organic matter pools with a specific rate of degradation, or by continuum models that seek to explore a range of oxidation rates (e.g. Arndt et al., 2013; Manzoni et al., 2009). In this section, we describe the different models of organic matter degradation used in our approach.

### 2.5.1 Oxidation models

There are no models that describe specifically the degradation of organic carbon within landslide deposits, but there are likely to be close analogies with those that are applied to soils and/or marine sediment (e.g. Arndt et al., 2013; Manzoni et al., 2009). The simplest model of the degradation of organic matter represents the carbon stock as one compartment, the so-called 'single pool' model (Berner, 1964). This pool is described by a mass of OC that loses carbon (during respiration the OC loss would be to $CO_2$) via first-order kinetics at a rate described by a single oxidation constant, $k_{ox}$ (% $yr^{-1}$). For a situation with no further OC input, such as a single landslide, the loss over time can be summarised as:

$$\frac{dM_{oc}}{dt} = -k_{ox} M_{oc} , \qquad\qquad (9)$$

An alternative is to describe the mass of OC by multiple compartments with different degradation rates and oxidation constants, referred to as the 'multi-pool model'. Here, the pools are treated in parallel, and the user defines the distribution of the initial OC mass between different pools. Each pool is described by an individual oxidation constant, which could conceptually relate to the reactivity of different organic compounds (e.g., Minderman, 1968) and/or the association of organic matter with different mineral phases (e.g. Hemingway et al., 2019; Mayer, 1994). This approach captures the observation that as organic matter degrades and ages, its apparent reactivity and rate of degradation decline (e.g. Middelburg, 1989).

In our study, a single pool model may be appropriate if the residence times of landslide deposits in the catchment are very short (annual timescales), so that it is only necessary to constrain the fate of the most reactive OC (Trumbore, 2000). This simple approach was applied by Wang et al. (2016) to draw conclusions on the fate of OC eroded by landslides during the Wenchuan earthquake. However, we know that at least some landslide deposits are likely to reside in the landscape for





decades or longer (Clark et al., 2016; Fan et al., 2018) and so a multi-pool approach may be more appropriate. In addition, landslides can mobilise the entire soil profile, including OC with different reactivity as a function of depth. For instance, the organic-rich O and A layers are highly reactive, whereas the mineral soil may have lower turnover rates (Tate et al., 1995), due to different organic compound reactivity and/or mineral-OC association (Hemingway et al., 2019). A multi-pool model

may also better capture processes operating within a landslide deposit. Burial of organic matter within the sediment pile, coupled with potential water-logging as water flow paths focus runoff from landslide scars into deposits (Emberson et al., 2016a; Lo et al., 2012), could mean that OC on the surface of deposits is more quickly degraded than OC at depth.

To constrain appropriate values of $k_{ox}$, we note that in New Zealand temperate forests, the mean OC turnover has been estimated to be ~20 years ($k_{ox}$ ~5 % yr$^{-1}$) for the top soil layers and $k_{ox}$ values of between 1.2% yr$^{-1}$ and 0.5% yr$^{-1}$ for the

total soil carbon (Tate et al., 1995). Thus, for the single pool model, we explore $k_{ox}$ values from 0.1% yr$^{-1}$ to 2% yr$^{-1}$ to reflect the total soil layer. For the multi-pool model, we assign three pools associated $k_{ox}$ of 2% yr$^{-1}$, 1% yr$^{-1}$ and 0.5% yr$^{-1}$ to reflect the local empirical constraint on soil OC degradation (Tate et al., 1995).

**2.5.2 Quantification of OC redistribution**

In our modelling approach, the temporal evolution of the initial mass of mobilized OC is described by a first-order oxidation

reaction law of the form (Blair & Aller, 2012; Trumbore, 2000; Wang et al., 2016):

$$M_{oc,t} = M_{oc,0}\, e^{-k_{ox}t}, \tag{10}$$

where $M_{oc,t}$ is the mass of OC contained in an individual landslide at each time step, $M_{oc,0}$ is the initial mass of OC mobilized by the landslide, and $k_{ox}$ is the first-order reaction kinetic constant for OC oxidation (Fig. 2a). While equation 10 only encompasses biochemical processes, the mobilized OC can also be physically exported from the deposit by fluvial

transport. In that case, the initial mass of mobilized OC synchronously decreases under the action of both biochemical degradation and fluvial transport as (Fig. 2a):

$$M_{oc,t} = M_{oc,0}\left(1 - \frac{t}{t_0}\right)e^{-k_{ox}t}, \tag{11}$$

The evolution of the mass of OC present in a landslide deposit therefore depends on its connectivity status. When a landslide deposit is not actively eroded by a river, the OC evolution is governed by equation (10). Once the landslide is connected to

the drainage network, the OC evolution follows equation (11) (Fig. 2b). The overall post-seismic OC export is then controlled by two timescales: i) the connection time of a landslide to the fluvial network, $t_{con}$; and ii) the export time of landslide fine sediment, $t_0$.





During the unconnected phase, the initial OC mass decreases only through oxidation and the overall mass of organic matter

that has been oxidised ($M_{ox,uncon}$) is estimated with the following equation:

$$M_{ox,uncon} = M_{oc,0}\, k_{ox} \int_0^{t_{con}} e^{-k_{ox}t}\, dt, \qquad (12)$$

In the case of a connected landslide, the organic matter contained within the deposit evolves as a function of the rate of

5   fluvial transport and oxidation. In order to quantify the OC redistribution, we compute the mass of OC exported by rivers (

$M_{riv}$), oxidised ($M_{ox}$) or remaining on the hillslopes ($M_{land}$) using this set of equations:

$$M_{riv} = \frac{\left(M_{oc,0} - M_{ox,uncon}\right)}{t_0} \int_0^{t_0} e^{-k_{ox}t}\, dt,$$

(13)

$$M_{ox} = M_{ox,uncon} + \left(M|oc,0 - M_{ox,uncon}\right) k_{ox} \int_0^{t_0} \left(1 - \frac{t}{t_0}\right) e^{-k_{ox}t}\, dt,$$

10   (14)

$$M_{land} = M_{oc,0} - M_{riv} - M_{ox}, \qquad (15)$$

Equations 8 to 13 describe the theoretical framework that we use to quantify the OC redistribution during post-seismic

periods. They are applied to several different oxidation models that we describe hereafter. The evolution of the OC stock in

each pool is managed by equations 10 (unconnected phase) and 11 (connection phase) to each landslide deposit.

15   **3 OC evolution of a single landslide deposit**

Before tackling the widespread mobilisation of OC by a large landslide population and coupled fluvial transport, we first

focus on theoretical predictions of OC evolution from a single landslide. This section focuses on the effect of the different

oxidation models and their associated oxidation constants ($k_{ox}$), and on the role of the connection time ($t_{con}$) and export time

($t_0$) in the partitioning of mobilized OC between oxidation and fluvial export (Fig. 3).

20   The first case (Fig. 3a) shows the evolution of the post-seismic OC content of a landslide deposit using a single pool

oxidation model for different values of $t_0$, $t_{con}$ and $k_{ox}$. The general tendency is that the proportion of OC exported by

riversdecreases with $t_0$, as longer export times enable more oxidation to take place. The latter effect is strengthened when the

landslide deposit undergoes a prior phase of disconnection from the river network. For instance, in the case $k_{ox}$ = 2% yr$^{-1}$,

when the landslide is directly connected to the river network, the quantity of the eroded OC transported by rivers is ~99%

Earth **Surface**
**Dynamics**
Discussions

when the export time is very short ($t_0$ = 1 yr), and decreases to 43% when the export time is long ($t_0$ = 100 yr). In contrast, when the deposit experiences a disconnection phase of 50 years, these proportions decrease to 36% ($t_0$= 1 yr) and 16% ($t_0$ = 100 yr). The choice of the oxidation constant value also plays a role in the fate of OC: for $t_0$ = 100 yr the proportion of OC exported by rivers is 79% for $k_{ox}$ = 0.5 %yr$^{-1}$, whereas it is only 43% for $k_{ox}$ = 2% yr$^{-1}$.

The multi-pool oxidation model reveals similar patterns as the single-pool model, with the mass of OC exported by the fluvial network decreasing with increasing values of $t_0$ (Fig. 3b). However, the inclusion of OC pools with lower reactivity results in more OC being exported by fluvial transport. When the low-reactivity pool is the largest, up to 75% of OC is exported by rivers even for long export times ($t_0$ = 100 yr). In contrast, with a large high-reactivity pool, only 50% of OC is preserved long enough to be exported by rivers with the same value of $t_0$.

Drawing on observations from soil profiles from New Zealand (Harvey, 2019; Tonkin & Basher, 2001), we conceptualise that soils profiles in the West Coast catchments comprise: 1) a highly reactive O layer, which can constitute the first pool and represents ~10% of the total soil mass; 2) the A and E soil layers constitute the second pool and represent ~20% of the total mass; and 3) the last pool contains the low reactivity organic matter that is present in the deeper B and C layers and represents the remaining ~70% of the total soil mass. Therefore, this soil configuration, with a large slowly reacting pool, is

likely to preserve the OC in landslide deposits for long timescales, and therefore promote a carbon sink.

## 4 Timescales of erosion as a CO₂ source or sink

The single landslide scenario can be used to explore the timescales over which erosion acts as a $CO_2$ source or sink. For instance, immediately following a landslide-triggering event, degradation of the organic matter will release $CO_2$ to the atmosphere (Feng et al, 2009). Conceptual models suggest that this impacts the rate of $CO_2$ drawdown via vegetation re-

growth and soil re-establishment (Restrepo et al., 2009). However, over longer timescales, the OC transported by rivers has the potential to contribute to net storage, and once the forest has regrown, landslide-driven erosion represents a net $CO_2$ sink (Berhe et al., 2007).

To place some quantitative constraint on these timescales, we explore the fate of landslide-mobilized soil OC with the single-pool degradation model ($k_{ox}$ = 2% yr$^{-1}$) for a range of fine sediment export times, $t_0$ and landslide connection times

$t_{con}$. The processes acting as a $CO_2$ sink are soil-re-establishment on landslide scars and fluvial transport, under the assumption that the OC experiences minor oxidation during fluvial transport (Scheingross et al., 2019). To describe the evolution of soil OC on hillslopes following a landslide event, we assume that it tracks the regrowth of forests. This is supported by the observation that surface soil horizons from the western Southern Alps have a high radiocarbon content (with a "bomb" $^{14}$C signature) that suggest rapid OC turnover on timescales of less than 50 years (Frith et al., 2018; Horan et



al., 2017). As such, the evolution of the C stock of the OC-rich surface soil horizons (to 0.1 m depth) may quickly track any re-growth of vegetation that supplies the OC inputs.

To model above ground biomass (AGB, in Mg ha$^{-1}$) growth through time, we call on an empirical model defined for a set of global measurements (McMahon et al., 2010):

$$AGB = \beta_1 \left( \frac{t}{t+\theta} \right) \qquad (16)$$

where $\beta 1$ is the asymptotic maximum biomass that a stand can achieve, $t$ is time, and $\theta$ is the age at half-saturation of the function. This model is generally used to describe the recovery of AGB following a disturbance. Based on equation 16, the full re-establishment of pre-event OC stocks is likely to take ~150-200 years for above ground biomass (McMahon et al., 2010). If we define this as the timescale over which carbon stocks approach pre-disturbance levels, $t_{recovery}$, we can explore the net $CO_2$ exchange associated with landslide mobilized OC.

These calculations demonstrate how landslide erosion can be viewed as both a $CO_2$ source (Chambers et al., 2009; Feng 2009), and sink (Wang et al., 2016), depending on the timescales of consideration. Without fluvial transport, the competition between above ground biomass growth and recovery of the surface organic soil horizons versus OC degradation, controls the presence and duration of a transient period of $CO_2$ release and $CO_2$ drawdown (Fig. 4a). A faster rate of soil re-establishment allows for the C stock on hillsides to build up faster than the oxidation of eroded soil OC, and the erosion event initially looks like a small sink. In contrast, slower soil recovery is associated with a period of $CO_2$ release, whose extent depends on the value of $\theta$. However, by the nature of the model set up, the net effect over the duration of a seismic cycle (i.e. ~270 years) is no net $CO_2$ drawdown; the landslide mobilized OC is oxidised, while the landslide scar builds up an equivalent new OC stock.

By including river sediment transport, the model produces a net long-term sink of $CO_2$ (Fig. 4b). In this case, a $CO_2$ source period only exists if the connection phase is long enough to allow for OC oxidation to play a significant role (Fig. 4b). For instance, the initial $CO_2$ source lasts 85 years in the case of $t_{con}$ = 50 yr, but lasts <10 years when $t_{con}$ = 10 yr. The size of the longer term $CO_2$ sink depends on the river transport efficiency, $t_0$ and on the connection time $t_{con}$.

In summary, if $t_{recovery}$ is ~200 years, it suggests that landslide erosion results in a short-term, transient source of $CO_2$ for less than ~50 years, and that the longer-term $CO_2$ sink persists over timescales >200 years. It is important to note that the AGB model applied here does not capture the formation of deeper, mineral associated soil OC, which may take longer periods of time and could contribute importantly to soil OC stocks (Harvey, 2019). Nevertheless, all the cases explored with our approach show that, over the ~270 year duration of a seismic cycle in this setting, landslides always act as carbon sink.



## 5 The fate of OC mobilized by a landslide population

The view from a single landslide deposit allows us to assess the fate of eroded OC for different configurations of landslide connectivity, fluvial transport efficiency, and rates of OC oxidation (Figs. 2 and 3). However, for a distribution of landslides triggered by an earthquake, we expect these to vary due to the spatial variability of river transport capacity (i.e. upstream

drainage area upstream of a landslide deposit), the timescale of deposit connection, and varied landslide areas and volumes. In this section, we explore what this complexity means for the fate of mobilized OC, considering a large landslide population triggered by a single earthquake.

Using the Quakos framework (Croissant et al., 2019), we model a $M_w$ 7.9 earthquake on the Alpine Fault that is consistent with paleoseismic reconstructions (Howarth et al., 2012, 2018). Our modelling approach predicts that ~58,000 landslides

would be triggered by the earthquake, impacting an area of ~11,000 km$^2$ and mobilising ~ 1.2 km$^3$ of fine sediment. Overall, the mass of OC mobilized from soil organic matter for this scenario is ~ 3.3 MtC. From this starting point, this section examines the role of various parameters on the post-seismic redistribution of OC. We denote the cumulative timescale necessary to evacuate fully the landslide mobilized OC as $t_{tot} = t_{con} + t_0$. The OC evolution is tracked over one seismic cycle of a duration of 270 years.

### 5.1 The role of landslide connectivity

The spatial variation of the local transport rate ($\overline{Q_s}$), combined with a landslide volume ($V_{ls}$) population, leads to the emergence of a distribution of landslide export times $t_0$ and of distances between deposits and the fluvial network. This in turn controls the connection time $t_{con}$. Here, we explore the fate of OC as a function of the connection velocity of landslides to the river network ($u_{con}$), which also controls the distribution of $t_{con}$.

For each individual landslide generated by Quakos, we calculate the proportion of OC exported by rivers as opposed to that oxidised to $CO_2$ as a function of $t_{tot} = t_{con} + t_0$. The general pattern that emerges is that the proportion of OC exported by rivers decreases with $t_{tot}$, ranging from 100% of OC exported by rivers for values of $t_{tot}$ < 1 yr, to 0% for $t_{tot}$ > 300 to 10,000 yr depending on the value of $t_{con}$ (Fig. 5). This approach allows us to assess an envelope of possible values of the proportion of OC exported by rivers as a function of $t_{tot}$, defined by two cases: i) when $t_{con}$ is large compared to the export

time (red line in Figure 5); and ii) when the landslides are directly connected to the rivers when $t_{con} = 0$ (blue line in Figure 5). The scatter in the data emerges from the competition between the two timescales controlling the fate of the landslide mobilised OC. A dominance of $t_{con}$ promotes the oxidation (and/or storage on hillslopes) of the organic matter, while a lower value of $t_0$ leads to a higher proportion of OC that is exported by the fluvial network. Thus, for a fixed climate (mean



annual runoff = 1 m yr$^{-1}$ and runoff variability $k$ = 1, in equation 7) the quantity of OC that is exported by the fluvial network is sensitive to the chosen connection velocity of the landslides to the rivers (Fig. 5).

We next investigate the impacts of the connection velocity on the ultimate redistribution of OC at the end of a seismic cycle. The fate of carbon is considered as either being returned to the atmosphere as $CO_2$, retained in the landscape (present in

landslide deposits), or contributes to exported river sediments (a potential for longer-term OC burial in lakes or oceans). Not surprisingly, the results for the single pool OC degradation model show that a lower connection velocity and a higher oxidation constant produce a larger proportion of OC oxidised and released in the atmosphere (Fig. 6). Interestingly, scenarios that show a significant storage of OC at the end of a seismic cycle only emerge for a low oxidation constant coupled to a slow connection velocity (Fig. 6). The multi-pool degradation model outputs do not differ significantly from the

single pool results. However, predicted OC storage in the mountain landscape is larger when the low-reactivity pool is the largest (Fig. 7). For both oxidation models, high connection velocities (> 100 m yr$^{-1}$) systematically promote the river transport of OC, with between 80 to 98% of OC being exported out of the catchments. The physical meaning of the connection velocity for the case of fine sediment is discussed in Section 7.1.

By considering a full landslide population, we can also consider the effect of landslide size on the fate of OC. Our results

suggest that intermediate landslides (i.e. with an area ranging from $10^3$ to $10^5$ m$^2$) are responsible for most of the ultimate export of river carbon (Fig. 8). While smaller landslides ($A_{ls} < 10^3$ m$^2$) are removed efficiently by rivers, they mobilize a lower initial quantity of OC and therefore are less important on the total carbon budget. In contrast, large landslides ($A_{ls} > 10^5$ m$^2$) individually mobilize large quantities of OC, but have longer export times as $t_0$ is a function of $V_{ls,fine}$, potentially reducing their impact on the carbon export by fluvial transport.

**5.2 The role of climate in landslide OC evacuation**

In our model, the mean annual runoff and runoff variability help determine the sediment transport rate (Equation 6 that depends on Equation 7) and therefore in the river export of OC. These factors have been highlighted as important for the transport of particulate OC in mountain rivers (Hilton, 2017; Wang et al., 2019). The fate of landslide-mobilized OC is sensitive to the export time, which is a ratio between the volume of fine sediment available for transport and the sediment

transport rate. Here, we assess how the fate of OC is moderated by these climatic parameters. For the mean annual runoff, we chose a range of values that span the observed values of different mountain ranges (e.g. Bookhagen & Burbank, 2010; Hicks et al., 2011). For the runoff variability, $k$ varies between 0.5, a value implying a high recurrence of large daily runoff event typical of Taiwan (Lague et al, 2005), and 4, a value representative of temperate regions that rarely experience flooding events (Croissant et al., 2019).

The role of runoff and runoff variability can be explored for different scenarios of the landslide dynamic connectivity, ranging from a slow connection velocity ($u_{con}$ = 1 m yr$^{-1}$) to the 'full connectivity' case. Generally, the proportion of OC





exported by rivers increases for higher mean annual runoff and higher runoff variability (i.e. higher frequency of large daily discharge, low value of $k$) (Fig. 9). For mean annual runoff values below 1 m yr$^{-1}$, the proportion of OC export by rivers is not significantly affected by runoff variability. In general, the results show more sensitivity to the mean annual runoff than to runoff variability. This emerges from the parameterisation of the long-term fluvial transport capacity (equations 6 and 7).

The suspended sediment transport law (equation 5) does not have a threshold, which limits the role of large runoff events on the long-term transport capacity. Therefore, the long-term transport capacity is controlled by intermediate runoff events which are set by the mean annual runoff value (see further explanation in Croissant et al., 2019).

The connection velocity has a major impact on the modelled fate of OC. However, climate strongly moderates this response. For instance, the full connectivity scenario shows that the proportion of the landslide mobilized OC that is exported by rivers

ranges from 30% for low runoff and low discharge variability, to ~95% for high runoff and variability (Fig. 9c). These model outputs predict that most feasible combinations of the climate parameters lead to a dominance of the fluvial export of OC. The low-connection velocity produces a specific response (Fig. 9a). This is because the amount of exported material is limited to only the initially connected landslides (e.g. ~30% in these simulations). We discuss the broader implications of these results in Section 7.

**6 Calculating long-term fluxes – application to the Southern Alps**

In the previous section, OC partitioning between the different reservoirs was expressed as percentages rather than absolute values to illustrate the flexibility of the approach. In this section, we apply the model to the New Zealand case for which we can constrain parameters using empirical datasets. In addition, we consider an upscaling approach, to quantify the fate of landslide-mobilised OC over several seismic cycles. We use the single pool oxidation model for simplicity ($k_{ox}$ = 1% yr$^{-1}$).

The parameterization of the tectonic and climatic forcing is chosen to mimic the West Coast of the Southern Alps. For instance, the mean annual runoff is set at 5 m yr$^{-1}$ and the runoff variability to $k$ = 1 (Croissant et al., 2017). The tectonic scenario consists of a series of M$_w$ 7.9 earthquakes occurring on the Alpine Fault every 270 years. We also explore three different landslide connectivity scenarios, ranging from full connectivity to $u_{con}$ = 1 m yr$^{-1}$.

Our approach models the pulsed, large-scale mobilisation of OC from earthquake-triggered landslides (Fig. 10). The OC can

be partitioned between that exported by rivers, and that released to the atmosphere as $CO_2$ by oxidation. For every scenario, earthquakes are identified by a pulse of OC exported by rivers that reaches a maximum in the year following the seismic event. The value of this peak is larger for the full connectivity scenario, which delivers a larger and immediate supply of sediment to the drainage network. After each earthquake, the exported OC decreases and tends towards zero at the end of each seismic cycle. The model shows the vast majority of OC export by rivers can happen during the first decade of the post-

seismic period. This is consistent with observations from a lake core downstream of a small catchment draining the Alpine Fault (Frith et al., 2018). In the model, the fast initial fluvial evacuation of OC is due to a large volume of sediment being initially connected to the fluvial network, combined with a climate that promotes high sediment transport rates. The amount



of OC is then exported at a lower rate over the next several decades. In the full connectivity scenario, the total volume of landslide-derived sediment is supplied to the rivers during the co-seismic phase. The decrease in the volume of landslide derived sediment is only controlled by the distribution of $t_0$ of the landslides as no new material is introduced in the fluvial network through time. In contrast, the two other scenarios include a progressive connection of new landslides to the river

network with time and therefore present higher OC export rates than the full connectivity case after ~10 years.

The amount of OC that is predicted to be oxidised is several orders of magnitude lower than the physical export of OC. However, oxidation becomes the dominant process after the initial large wave of fluvial export in the first decade that follows each earthquake (Fig. 10). In addition, oxidation fluxes are larger when the connection velocity decreases and river export becomes less efficient at removing OC.

Over one seismic cycle, the model approach suggests that the vast majority of the OC mobilised by earthquake-triggered landslides is exhausted from the landslide deposits for every scenario, i.e. there is no net accumulation of fine OC in the landslide deposits during the time span of 270 years. Once again, the landslide connection velocity influences the fate of OC, with ~90% of the OC exported by rivers for the full connectivity scenario, while this value decreases to ~28% for the slow connection velocity case. It is worth noting that without any sediment transport, the single pool degradation model would

predict that ~89% of the organic carbon would have been oxidised and returned to the atmosphere and the 11% of remaining OC stored in landslide deposits over timescales longer than the seismic cycle.

# 7 Discussion

## 7.1 Comparison to previous work

Previous work has sort to quantify the provenance and flux of particulate OC following large storm events based on river
gauging station data (Clark et al., 2017; Hilton et al., 2008), remote sensing approaches (Clark et al., 2016; Ramos Scharrón et al., 2012; West et al., 2011), or lake stratigraphic records (Frith et al., 2018; Wang et al., 2020). These studies provide insights on the rates and processes of OC transport to the oceans (Galy et al., 2015; Hilton 2017). However, they are generally unable to constrain the roles of OC oxidation in the landscape (Fig. 2) and above ground biomass re-growth (Fig. 4) in the OC budget of a landscape recovering from these damaging events. In our study, we propose a theoretical
description of these processes using a simplified numerical approach.

The export of OC during the post-seismic period depends strongly on the fine sediment export rate (Hovius et al., 2011; Tolorza et al., 2019; Wang et al., 2015; Wang et al., 2016). A few examples have shown that, in wet climates, suspended load fluxes after a large earthquake are characterized by a rapid increase directly after the seismic event, which is sustained for less than a decade, before returning to background levels (Hovius et al., 2011). This behaviour is reproduced by our
simulations (Fig. 10). However, this does not imply that all the fine sediment has been evacuated out of the epicentral area. For instance, abundant landslide deposits located in headwater areas (Ramos Scharrón et al., 2012), or fine particles trapped



under coarser grains in the deposits, would slow down the evacuation of sediment once the easily accessible particles have been entrained. In that regard, the scenarios that we propose in Section 6 show possible trajectories of the partitioning between oxidation and fluvial export, with oxidation being the dominant process after an initial pulse of fluvial evacuation of OC. However, we note that the initial pulse is present in all simulations, independent of the connection velocity, and implies

that in all cases fluvial export is the dominant fate of OC mobilized by landslides in a setting like the Southern Alps. These findings are consistent with the limited available measurements made following the 2008 Wenchuan earthquake (Wang et al., 2016) and fit the pattern of organic matter accumulation seen in lake deposits from the western Southern Alps that record four past Alpine Fault earthquakes (Frith et al., 2018).

Other research has sought to assess the role of landslide erosion as a $CO_2$ source or sink. For instance, Ramos Scharrón et al.

[2012] proposed a post-landsliding OC evolution model that accounts for the competition between carbon gains and losses. Their modelling approach differs from ours, because the processes that result in a $CO_2$ sink only include soil formation, above ground biomass growth on landslide scars and OC storage in landslide deposits. In their approach, fluvial transport contributes to oxidation as part of the OC source term. Despite these important differences, their results also suggest that widespread mass wasting is a $CO_2$ sink over long timescales in settings where landslide deposits have long residence times

(centuries), which would correspond to our low connection velocity scenario. Our results differ from theirs in that we model connection of landslides to the mountain river network, leading to scenarios in which up to 100% of the OC mobilized by landslides is removed by fluvial export rather than oxidised in the landscape.

## 7.2 Extreme erosion events and the global OC cycle

The mobilisation of OC by events that trigger widespread landsliding can produce large pulses of carbon export from the

biosphere (Clark et al., 2016; Frith et al., 2018; Wang et al., 2016; West et al., 2011). These carbon fluxes are large enough to raises questions about the role of large erosive events associated with earthquakes and storms in the long-term, or geological, carbon cycle (Hilton and West 2020). In this study, we propose a framework to examine the fate of OC eroded during a large earthquake and isolate how the climatic context plays a role in promoting fluvial transport. We do not attempt to quantify any subsequent degradation of OC in the marine realm, or within sedimentary deposits. In the context of 'short'

mountain ranges along active continental margins, with sediment transport distances of <100 km and rivers that are coupled to deep ocean sedimentary sinks (e.g., Mountjoy et al., 2018), it has been shown that 70% or more of the OC exported in river sediments is buried in offshore sedimentary deposits, and therefore is considered a long-term sink (Blair & Aller, 2012; Galy et al., 2015; Kao et al., 2014).

Our modelling exercise has shown that earthquakes can contribute to long-term, biospheric carbon sinks in the Southern

Alps case, with minimal within-catchment oxidation of eroded OC (Fig. 10). The model is broadly relevant to other mountain catchments in the western Pacific: Taiwan, and Papua New Guinea ( Dadson et al., 2005; Ferguson et al., 2011; Hicks et al., 2011; Hilton et al 2008). However, these settings also share geomorphic, tectonic and climatic attributes with islands of the Caribbean (e.g. Allemand et al., 2014) and of Central America (e.g. Ramos Scharrón et al., 2012), and some





catchments draining the western American continent (Leithold et al., 2006). In these conditions, the combination of short sediment travel distance and limited potential for fluvial sediment storage may promote efficient carbon transport from land through river systems, with minimal loss from the particulate OC pool (Scheingross et al., 2019). However, a complete understanding of the role of earthquakes on the carbon cycle has to also encompass catchments with large floodplains, such

as the large river systems that drain the Himalayas and the Andes. For instance, Galy et al., 2008 showed evidence for the replacement of Himalayan biospheric OC in the Ganges floodplain, and particulate OC eroded from the Andes is thought to suffer a similar fate during transport through the Amazon floodplain rivers (Bouchez et al., 2014; Ponton et al., 2014). We must also consider drier climates, where landslide OC stocks in soils may be lower and landslide debris may reside in catchments for much longer periods of time.

Our model has been applied to a single case study, with a specific tectonic and climatic regime (Fig. 10). It thus represents a first step in the understanding of the role played by widespread landslide-triggering events on the global carbon cycle. Ideally, future work can explore the diversity of mountain ranges that present different vegetation and soil cover, topography and climate, with a specific consideration of the recurrence time of earthquakes (Fan et al., 2018). Additionally, a complete carbon budget (Hilton & West, 2020) could be assessed if other long-term $CO_2$ sinks via silicate mineral weathering that

occurs in landslide deposits are considered (Emberson et al., 2017), alongside $CO_2$ release from sulfide mineral oxidation (Emberson et al., 2016a, 2016b) and oxidation of rock OC (Hilton et al., 2014).

**7.3 Limitations of the approach**

In this paper, we couple the reduced complexity model Quakos to organic matter degradation models to assess the fate of OC following a large landslide triggering event. As with any modelling approach, our model contains some limitations that are

discussed here.

Prediction of the landslide pattern for an event that has not taken place during the period of modern instrumental records is a difficult endeavour. We were able to constrain the shape of the landslide distribution using observations of rainfall-triggered landslides in the Southern Alps (Hovius et al., 1997). Nevertheless, uncertainties remain as to the landslide spatial density that would emerge from a $M_w$ ~8 earthquake on the Alpine Fault (Robinson & Davies, 2013). While this parameter does not

significantly affect the amount of OC redistributed in the different reservoirs (Fig. 6), it would greatly impact the OC fluxes and therefore their comparison to background fluxes. Our computations show that for the 'full connectivity' scenario, maximum post-seismic OC fluxes derived from the fluvial export would correspond to ~2 times (for a peak density of 4%) to ~6.5 times (for a peak density of 8%) the background fluxes computed from Hilton et al. [2008].

One of the main limitations of our modelling approach is in the assumption of a connection velocity $u_{con}$. As discussed in

Croissant et al. [2019], the lack of a theoretical framework describing the evolution of non-cohesive material sitting on hillslopes after a landslide-triggering event makes the description of this process difficult. For fine particles, there are several key processes that need to be considered across a range of scales. These include those operating at the grain scale (e.g.,



creep, rainsplash) to those operating over length scales of meters (e.g., dry ravelling, runoff driven erosion) to 100s of meters (subsequent landsliding, debris flow). There is also the fact that the availability of fine sediment is likely to change over time, for instance through the initial removal of easily transportable debris (Wang et al., 2015; West et al., 2014; Zhang et al., 2019). To move more finer material, subsequent mass movements may be needed to expose it within deposits, although

the distribution of fine material in landslide deposits also remains poorly constrained (Fan et al., 2019). It is tempting to compare the values of connection velocity chosen here (1 to 100 m yr$^{-1}$, or 3 x 10$^{-8}$ to 3 x 10$^{-6}$ m s$^{-1}$) to the velocity of water flow sediment export via a range of processes (e.g. DiBiase et al., 2017). In this case, the chosen connection velocities are very slow. However, they seek to describe the net motion of a large sediment pile, and as such, seek to integrate a range of processes that occur discretely in space and time.

Numerous physically-based and empirically-based models have been proposed to explore and predict hillslope transport processes, and some of these better capture the complexities of the processes known to operate (Aksoy & Kavvas, 2005; DiBiase et al., 2017; Tucker & Bradley, 2010). Future work could seek to incorporate more sophisticated transport laws into the framework we describe here. This would also allow the links between landslide re-mobilisation and climate to be explored, as there is likely to be a strong link between movement of existing landslide debris and events with high rainfall

intensity and high river stage (Marc et al., 2015). Useful insights could be gained using morphodynamic models with a realistic grain size distribution (Fan et al., 2019). However, perhaps the largest source of uncertainty relates to the dynamics of particulate OC transport versus clastic sediment in relation to its mobility following landslide events. While the general link between the two phases has been demonstrated (Galy et al., 2015; Hilton et al., 2012), there is also evidence that discrete clasts of woody debris can be sorted by flowing water (Hilton et al., 2015; Lee et al., 2019), and its transport behaviour

different from the clastic load due to a lower density and different particle shape (Turowski et al., 2016).

In our simulations, we simplify how the degradation of OC proceeds following its mobilisation of soil from hillslopes. First, it is possible that the landsliding process sorts and mixes the organic matter in such a way to promote oxidation of some of the materials, but to protect some from degradation. For instance, observations in the field suggest that large woody debris may concentrate near the surface and toe of the deposit after a landslide. Instead, finer material may mix deeper in the

deposit (Hilton et al, 2008). If the landslide deposit has low porosity, and/or receives runoff focused from the landslide scar (Emberson et al., 2016a), water-logging may promote preservation of this material under anoxic conditions until it is eroded and entrained by water. This would act to enhance the proportion of OC exported by erosion processes. Second, we do not account for OC contained within sedimentary bedrock. In the western Southern Alps, the OC content of rocks is low (~0.15%), but oxidative weathering is thought to be enhanced by high erosion rates (Hilton et al., 2014; Horan et al., 2017).

In the future, improved constraint on the reactivity of rock-derived OC (Hemingway et al., 2018) and models that link the production of fine clastic sediment to weathering reactions (Carretier et al., 2018) could be used to explore the role of large landslide populations on this mechanism for CO$_2$ release.

Finally, we assume there is no OC oxidation during river transport. Our New Zealand case study is focused on short rivers, with maximum distances from headwaters to oceans of less than a hundred kilometres. This assumption is thus supported by





recent work by Scheingross et al. [2019] who showed that within-river oxidation would be minimal, with between 0 to 10% of mobilized OC being oxidised during transport over a thousand kilometres. We also note that in the simulations, intermediate discharge events dominate the long-term transport rates (Croissant et al., 2019), meaning that most of the OC would be exported by flows that do not escape the river channel, and thus not go overbank onto the floodplains. Although, this dynamic could be altered if a sediment bedload wave modifies significantly the river bed elevation (e.g. Hancox et al., 2005; Korup, 2004). Finally, we recognise that in the Amazon River floodplain, high altitude derived OC from mountain catchments is thought to be lost/overprinted by lowland OC (e.g. Bouchez et al., 2014; Ponton et al., 2014). This means the geomorphic configuration of mountain catchments impacted by earthquake-events, could play an additional role in the fate of the mobilised OC.

## 8 Conclusions

In this study, we propose a new theoretical framework to quantify the fate of soil-derived OC mobilized by earthquake-triggered landslides. The approach combines an empirical model to compute a landslide population triggered by large earthquakes and a suspended sediment transport law with different models of OC degradation. Overall, this model allow us to quantify the fate of landslide-mobilized OC in terms of its potential contribution to atmospheric $CO_2$ (by respiration of OC), its transient storage in landslide deposits, or contribution to particulate OC in rivers (a potential $CO_2$ sink). The model is computationally efficient, and these features can be explored for a range of climatic forcing (in terms of mean annual runoff and runoff variability) and OC degradation rates, over multiple earthquake cycles.

At the scale of a single landslide, we find that the fate of OC is strongly controlled by the physical processes acting on the deposit. These include its initial connectivity to the river network and its potential to connect through time. This dynamic landslide connectivity is modelled using a "connection velocity" term, which seeks to capture a range of processes that may redistribute fine clastic sediment and OC from landslide deposits. The connection velocity and local river transport rate combine to set the time available for OC oxidation. The oxidation constants also play a significant role in the ultimate fate of OC, and a multi-pool degradation model can capture a persistence of OC throughout a seismic cycle of hundreds of years.

At the scale of an entire earthquake-triggered landslide population, the connection velocity and the mean annual runoff control the redistribution of OC. A wet climate and fast connection of fine material to the fluvial network both promote the riverine export of OC. Importantly, the type and complexity of the biogeochemical degradation model does not significantly affect the results. We apply the model to a case study setting in the western Southern Alps, New Zealand. The simulations suggest that much of the OC eroded from mountain ranges typical of the southwest Pacific would be efficiently transferred to the ocean during the first 1-10 years after a large earthquake. In this context, depending upon the fate of OC downstream, earthquakes are likely to promote net carbon sequestration. An extended application of this methodology tailored to other mountain ranges in different tectonic and climatic contexts would allow for a more precise determination of the role of earthquakes and widespread landslide events on the regional and global carbon cycles.



## Author contribution

T.C, G.L. and R.G.H designed the study. T.C., G.L., R.G.H and P.S. developed the theoretical description of the processes. J.W., E.H., and R.G.H. ran the quantification of the soil organic carbon content. T.C. analysed the data and interpreted them with inputs from R.G.H., J.H and A.D. T.C and R.G.H wrote the paper with inputs from all co-authors.

## Competing interests

The authors declare that they have no conflict of interest.

## Acknowledgments

This work was supported by a UK Natural Environment Research Council  Standard Grant awarded to R.G.H., A.L.D., and J.D.H (NE/P013538/1), and a Rutherford Foundation Postdoctoral Fellowship to J.D.H. (RFTGNS1201-PD). J.W. was supported by a COFUND Junior Research Fellowship at Durham University.

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



Earth **Surface**
**Dynamics**
Discussions

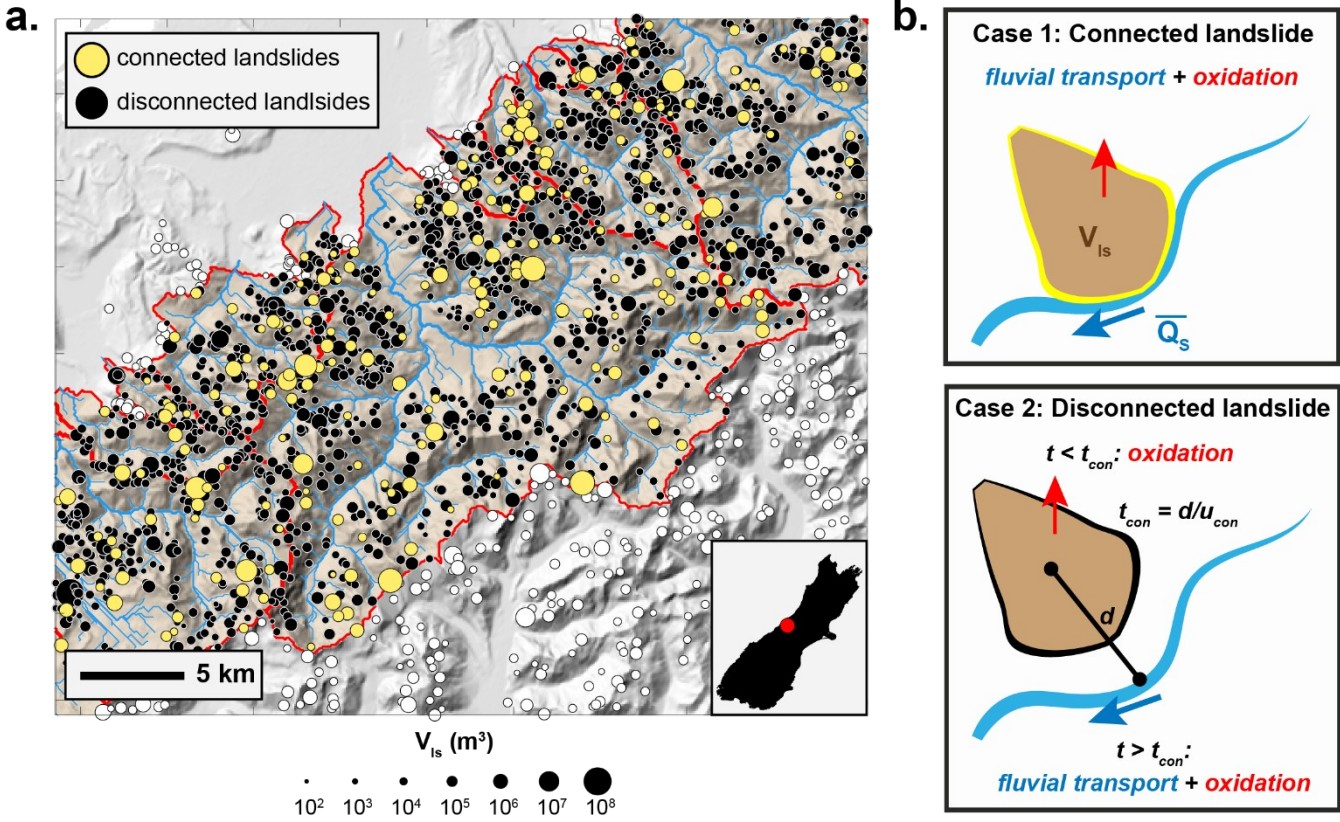

**Figure 1: a.** An example of a modelled distribution of earthquake-triggered landslides from the model QUAKOS. The view is centered on the Whataroa catchment in the West Coast of the South Island of New Zealand, The dot size is proportional to the landslide volume. Red
5 lines outline the some of the West Coast catchments that are accounted for in the simulations. The white dots are landslides generated by QUAKOS but not account for in our simulations. **b.** Illustration of the two possible connectivity status, and their connection velocity ($u_{con}$), distance from channel (d), and time for connection to the channel ($t_{con}$) for each individual landslide of volume $V_{ls}$.





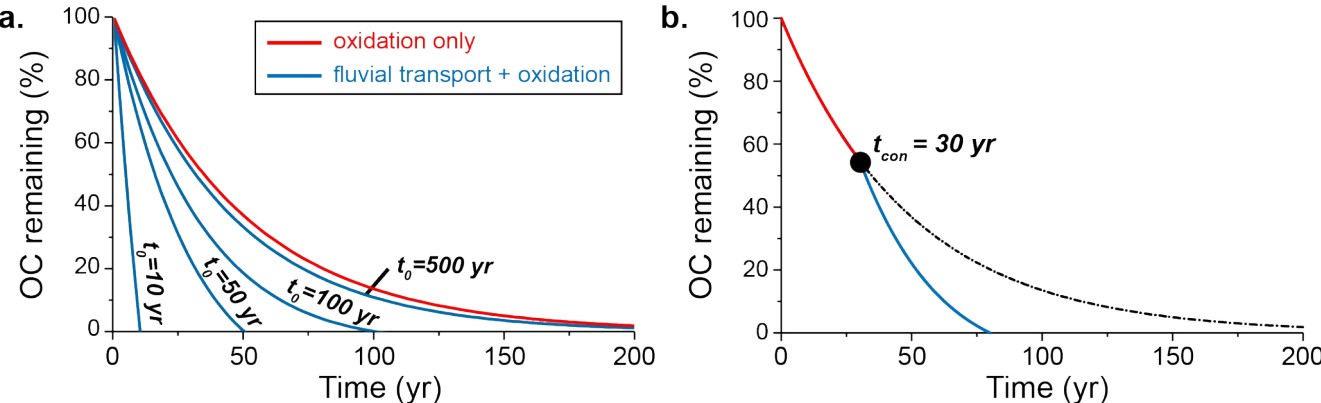

**Figure 2:** The fate of landslide-mobilized OC, considered in terms of the proportion of the mass of OC remaining within a single landslide deposit as a function of time. **a**. An example of a connected landslide, where the OC decrease is due to physical export as well as oxidation acting simultaneously (blue lines). The impact of the evacuation time is showed here by different value of $t_0$ (in years). The case of OC oxidation alone, with no physical export, is represented by the red line. **b**. An example of a landslide deposit that is initially disconnected from the river network and that connects after a connection time, $t_{con}$ = 30 yr, and is exported over a duration of $t_0$ = 50 yr. The dotted ride line represents the OC oxidation alone.



Earth **Surface**
**Dynamics**
Discussions


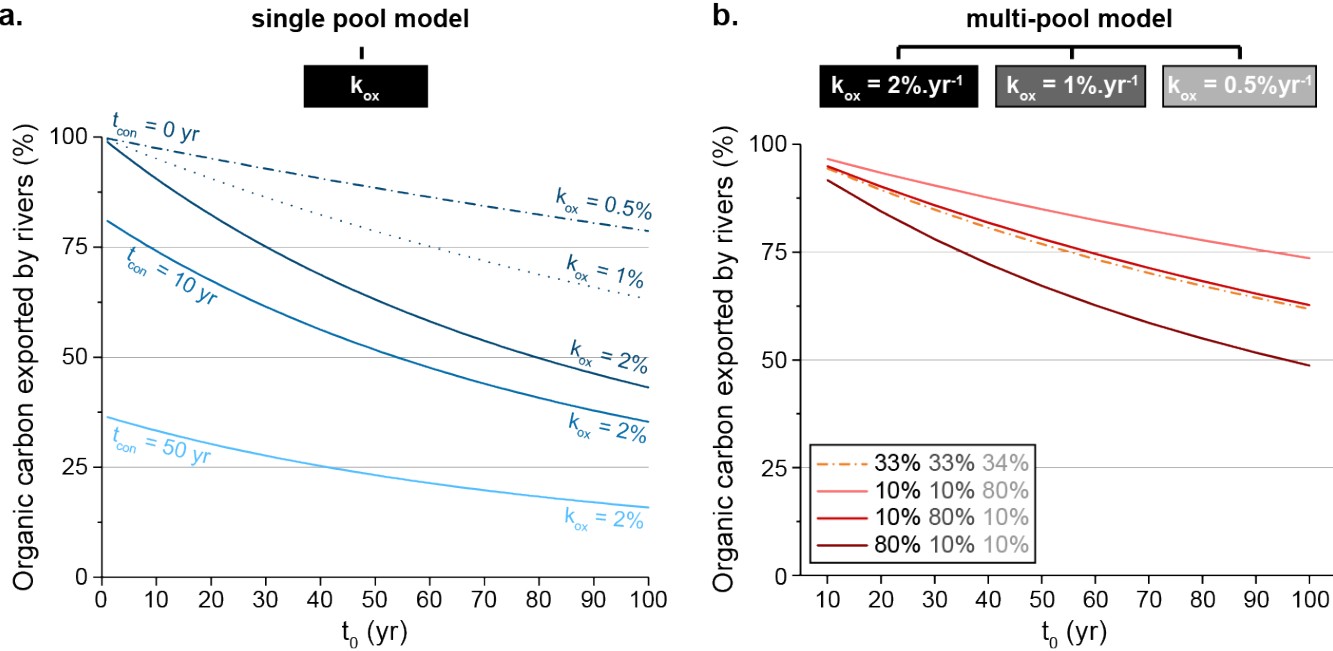

**Figure 3:** The proportion of landside-mobilized OC that is exported by rivers for a single landslide and different OC oxidation models, plotted as a function of the total export time ($t_0$). **a.** Results from the single pool model, using different values of the oxidation rate constant $k_{ox}$ = 0.5, 1 and 2 % $yr^{-1}$. Three different times of connection to the river network are also shown ($t_0$ = 0, 10 and 50 yr). **b.** Results from the multi pool model. Here the landslide is initially connected to the river ($t_{con}$ = 0 yr). The different lines represent model outcomes with different proportions of OC in the three modelled pools, as indicated by the greyscale text. Note that a larger fraction of OC in the high-reactivity pool leads to a decrease in the proportion of OC that can be exported by the river network.

Earth **Surface**
**Dynamics**
Discussions

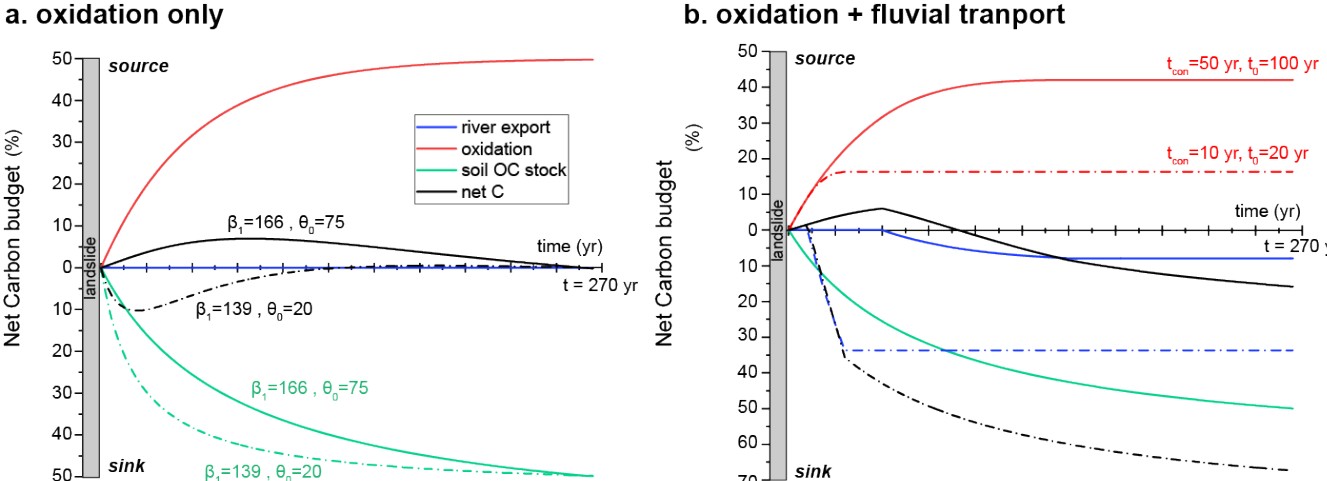

**Figure 4:** The modelled transient C budget of eroded organic carbon (partitioning between a $CO_2$ source or sink) over a seismic cycle of 270 years duration. Red lines are the $CO_2$ source from oxidation of the mobilized OC. Green lines indicate $CO_2$ sequestration by the re-establishment of the soil OC stocks in the landscape, which is based on a model of above ground biomass accumulation. **a.** The case where only OC oxidation is accounted for. The parameters controlling the shape of the soil OC stock model are explored, and determine whether landsliding is a net C source (solid black line) or net sink (solid dashed line). **b.** The case where OC oxidation and fluvial export are both active. Here, the parameters controlling the physical export of sediment ($t_0$ and $t_{con}$) are explored. Solid lines correspond to $t_0 = 100$ yr and $t_{con} = 50$ yr, and result in a short-term C source but long-term sequestration. Dashed lines correspond to $t_0 = 20$ yr and $t_{con} = 10$ yr, and result in net sequestration over the entire seismic cycle.



Earth **Surface**
**Dynamics**
Discussions

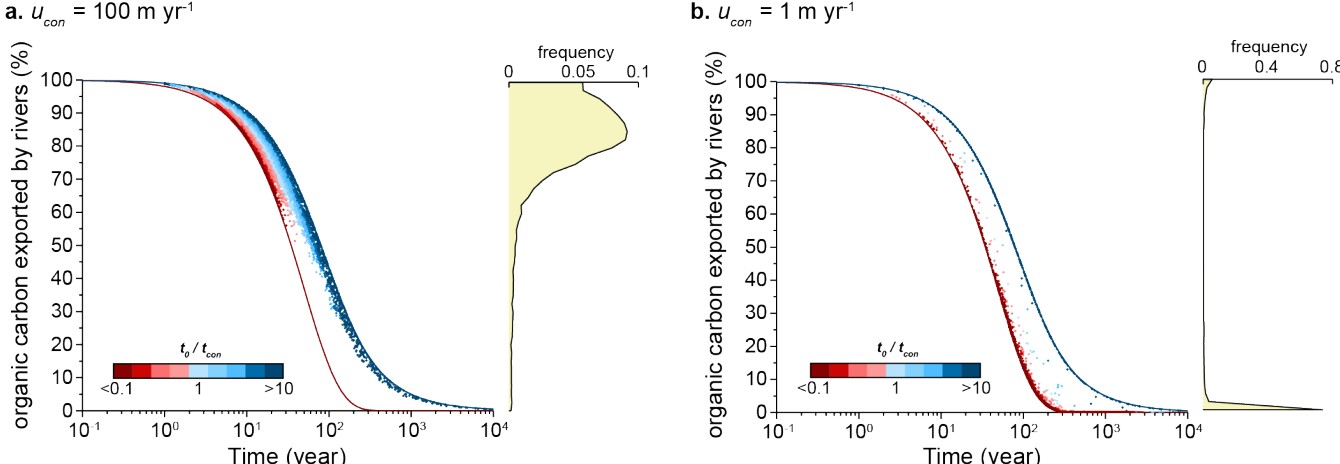

**Figure 5:** The proportion of landslide-mobilized OC that is exported by rivers for a population of co-seismic landslides, plotted as a function of the total time the landslides remain in the catchment. All results are for a single pool model of OC oxidation and different values of the connection velocity: **a.** $u_{con}$ = 100 m yr⁻¹, **b.** $u_{con}$ = 1 m yr⁻¹. Each dot is an individual landslide characterized by a combination of its connection time ($t_{con}$) and fluvial export time ($t_0$), and its position on the x-axis is $t_{tot} = t_{con} + t_0$. The colour of each dot represents the ratio between the export time and the connection time. The blue line is the percentage of OC exported by rivers when $t_{con}$ = 0. In this case, the x-axis corresponds to $t_0$ only. The red line represents the lower limit of the percentage of OC that could be exported by rivers for a given time. In this case, $t_0$ = 0 and the x-axis corresponds to $t_{con}$ only. The histograms on the right of each plot represent the distribution of the percentage of OC exported by rivers from the scatter plot.





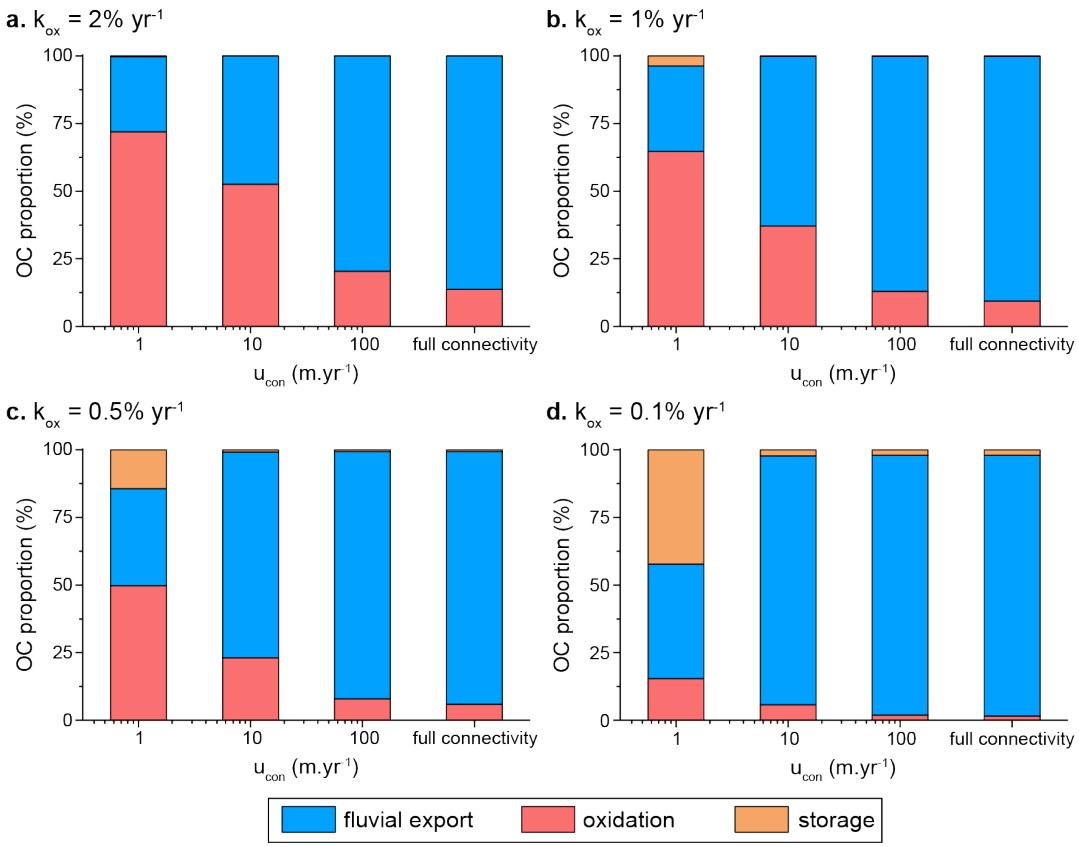

**Figure 6:** Histograms showing the fate of the landslide-mobilized OC at the end of a seismic cycle of 270 years duration, partitioned between fluvial export, oxidation, and landslide-deposit storage. These results have been computed with the single pool model for different connection velocities and different oxidation constants: a. $k_{ox}$ = 2% yr$^{-1}$; b. $k_{ox}$ = 1% yr$^{-1}$; c. $k_{ox}$ = 0.5% yr$^{-1}$; d. $k_{ox}$ = 0.1% yr$^{-1}$;

Earth **Surface**
**Dynamics**
Discussions

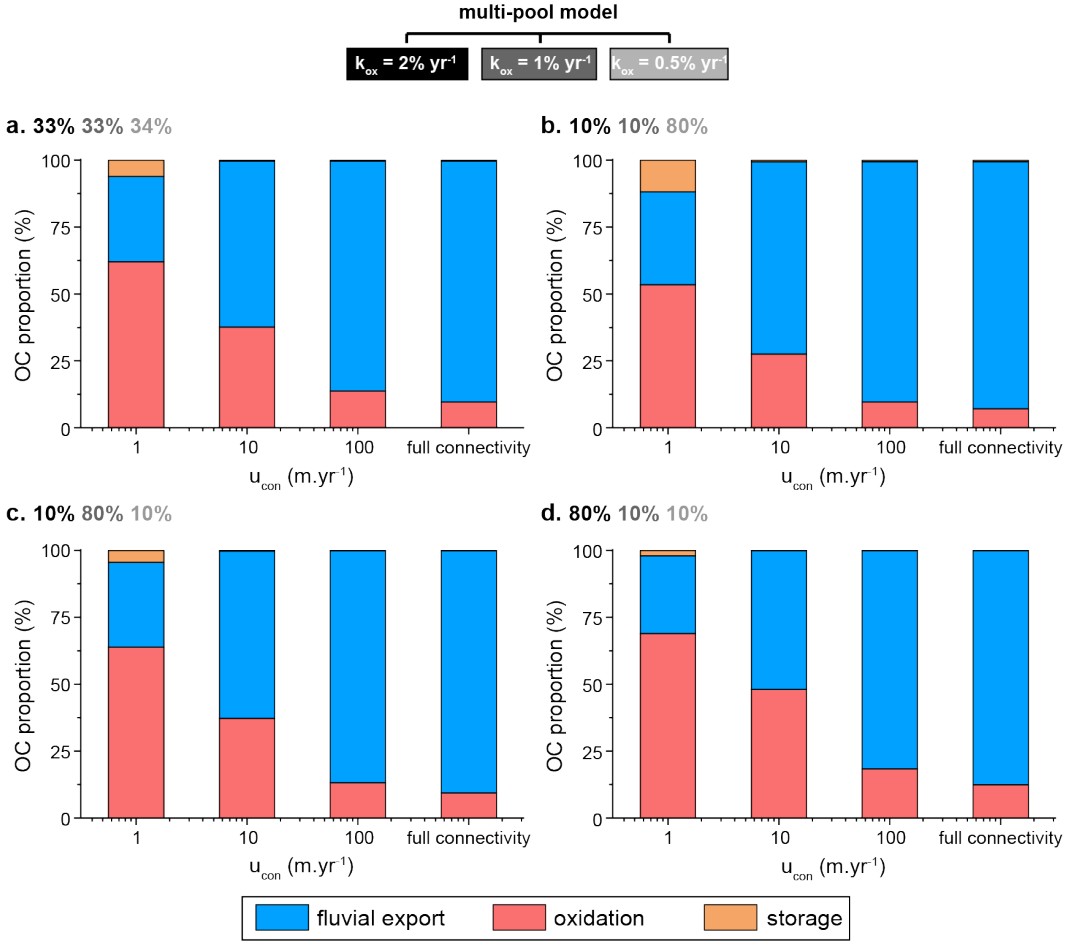

**Figure 7: :** Histograms showing the fate of the landslide-mobilized OC at the end of a seismic cycle of 270 years duration, partitioned between fluvial export, oxidation and hillslope storage. These results have been computed for different connection velocities with a multi-pool model composed of three pools with oxidation constants $k_{ox}$ = 2% yr$^{-1}$, 1% yr$^{-1}$ and 0.5% yr$^{-1}$. Panels (a)-(d) show model outcomes with different proportions of OC in the three modelled pools, as indicated by the greyscale text: a. equal distribution between pools b. dominance of the low reactivity pool. c. dominance of the intermediate reactivity pool. d. dominance of the high reactivity pool.





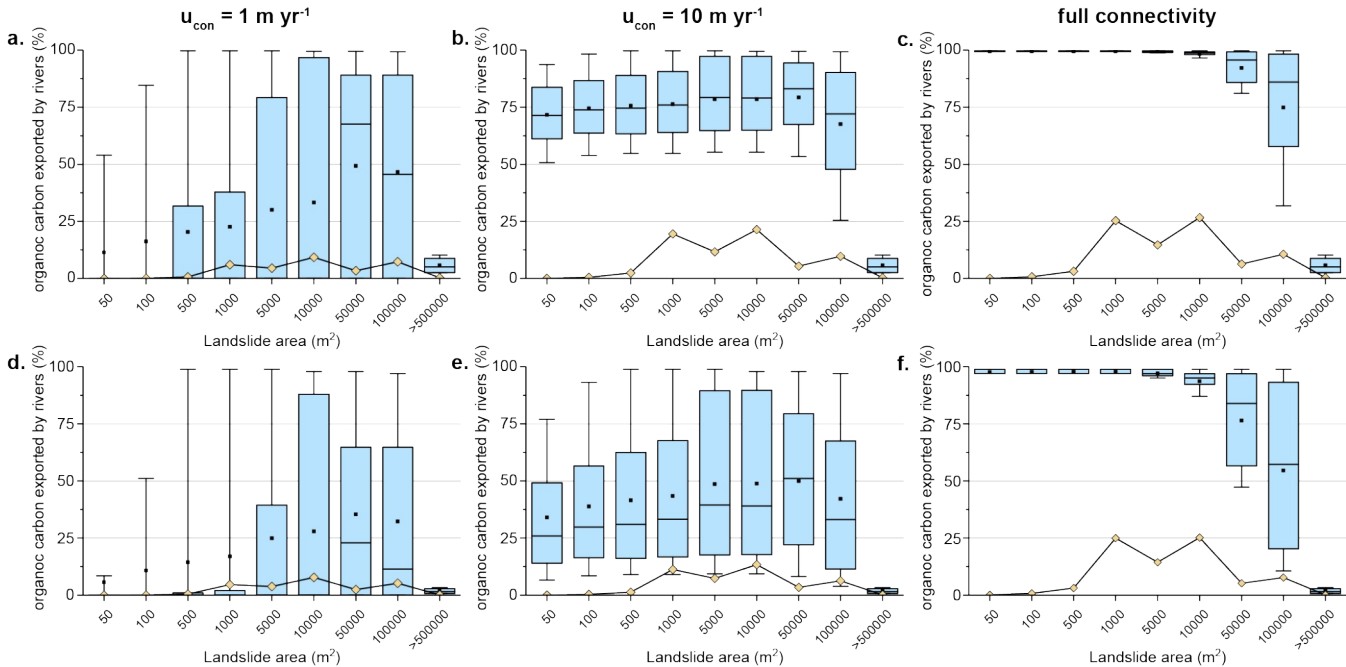

**Figure 8:** Quantification of the export of OC by river transport for different classes of landslide area (boxplots) and for different oxidation constants of the single-pool model (top row: $k_{ox}$ = 0.5% yr$^{-1}$; bottom row: $k_{ox}$ = 2% yr$^{-1}$). The yellow diamonds represent the contribution of each landslide area class to the total OC exported by rivers. The boxes show the 25$^{th}$-75$^{th}$ percentiles, the whiskers show the 10$^{th}$-90$^{th}$ percentiles, the dot shows the median, and the horizontal bar shows the mean.





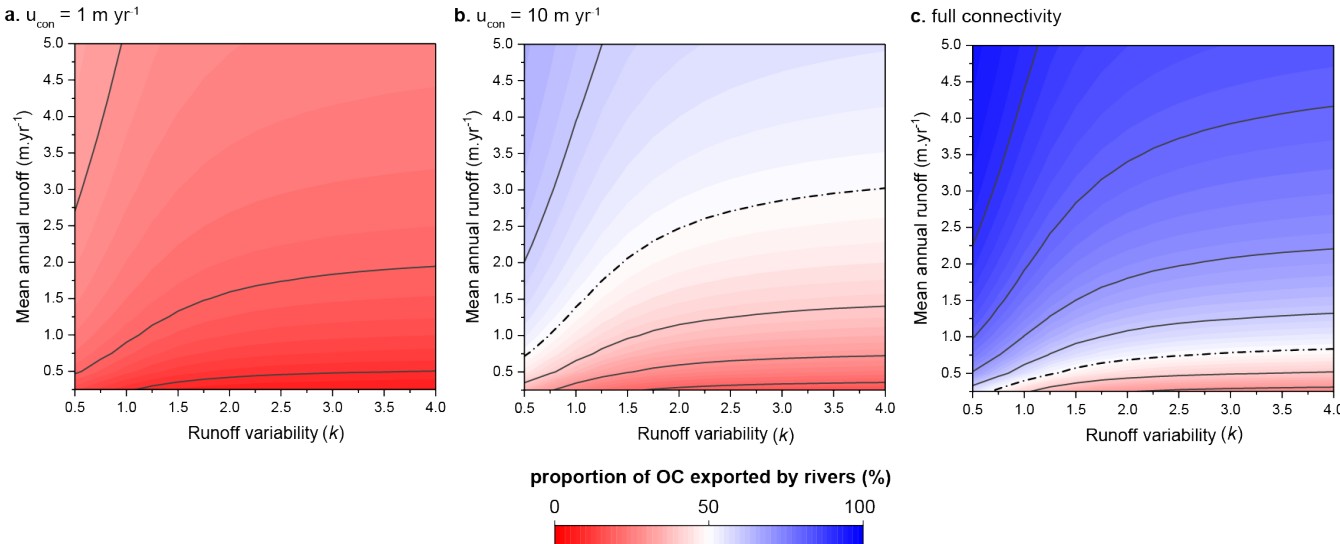

**Figure 9:** Proportion of landslide-mobilized OC that has been exported by rivers over one seismic cycle as a function of mean annual runoff and runoff variability. Three scenarios of landslide connection velocity are shown: **a**. $u_{con}$ = 1 m yr$^{-1}$, b. $u_{con}$ =10 m yr$^{-1}$ and c. full connectivity. The lines are representing contours of the proportion of OC exported by rivers and are incremented every 10% but the dashed line which represents the 50% contour.





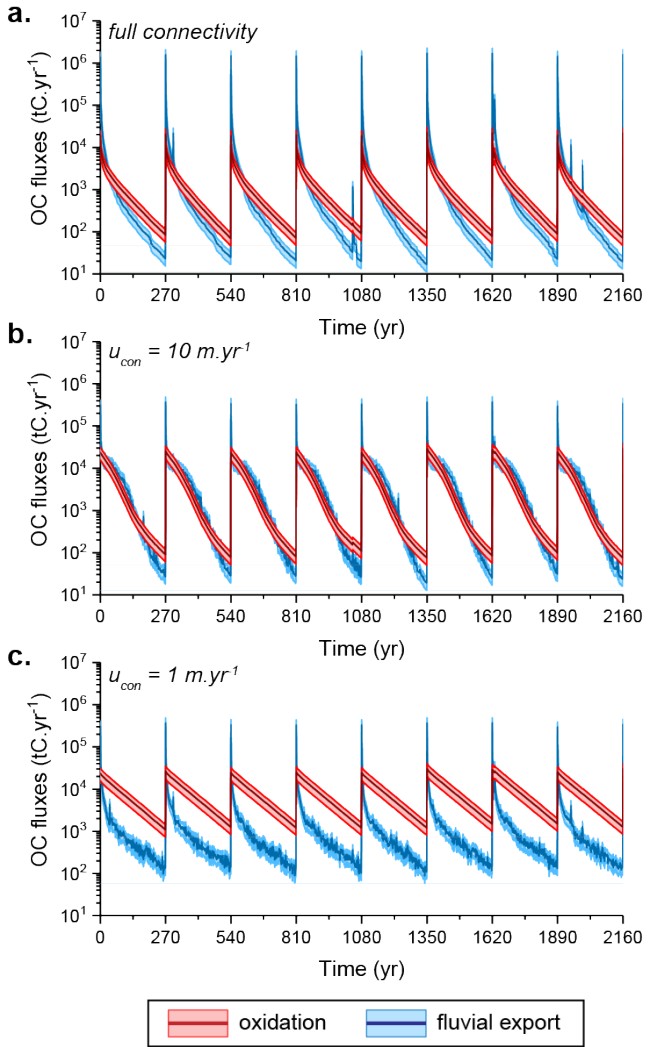

**Figure 10:** The modelled fate of landslide-mobilized OC over several seismic cycles, based on parameters set by the Alpine Fault, western Southern Alps, New Zealand, with the single pool model and $k_{ox}$ = 1 % yr$^{-1}$. Each panel shows the fluxes of OC exported by rivers (in blue) and the release of $CO_2$ to the atmosphere from OC oxidation (in red) for different connection velocity scenarios: a. full connectivity, b. $u_{con}$ = 10 m yr$^{-1}$, c. $u_{con}$ = 1 m yr$^{-1}$. The envelope for each plot accounts for the uncertainties in the soil OC content.