# Peer review of "Pulsed carbon export from mountains by earthquake-triggered landslides explored in a reduced-complexity model"

_Earth Surface Dynamics, 2020_

## Referee Comment (RC1) · Sebastien Carretier (Referee) · 21 Dec 2020

This manuscript is very well written and easy to read. It constitutes a real added value in the understanding of the CO2 budget of mountains with respect to organic carbon, taking into account the spatial and temporal variability of the competition between oxidation and physical export of OC at the scale of a catchment (landslides + rivers) and over a seismic cycle. The figures are self-explanatory. The limits of the model are clearly indicated and the conclusions are supported by the results. I think this contribution is close to being in good shape for publication.

In addition to the specific points listed below, I suggest considering the possibility of

presenting part of the results in an adimensionalised form. For example, according to the formalism of the model, it seems that an important parameter is the dimensionless product to * kox. I therefore wonder whether the curves in Figures 2 and 3 in particular would not collapse by grouping the curves by value of to * kox and (to+tcon) * kox.

Concerning to, the time of evacuation by rivers, Thomas Croissant has shown in another paper (Croissant et al., 2017, Nat Geo) that the export time of connected landslides no longer significantly depends on the size of the landslide above a threshold, thanks to the lateral dynamics of the rivers incising these landslides. In this contribution, Thomas Croissant et al. do not seem to consider this limitation. This saturation process may reinforce their conclusion of an effective export of OC and may deserve comment.

The study of a seismic cycle seems to consider only one magnitude of earthquakes (7.9) occurring at regular intervals. What would be the contribution of other possible earthquakes to the C balance?

Specific comments

P8L20 Define better what is a "pool" (landslide? Soil layer within a landslide?)

P8L25 Not clear to me how the multi-pool approach used here relates to this observation that kox should depend on age.

P8L27 (and P7L9 and P9L8) Residence time, turn over time etc: These "times" have a precise meaning in the Reservoir Theory underlying this work, and do not necessarily match. If the distribution of OC ages is heavy-tailed, the concept of Residence Time may be not sufficient to characterise the oxidation process or any process relating to the ageing of C (e.g. Mudd and Yoo, JGRES, 2010, https://doi.org/10.1029/2009JF001591). Could you comment on this?

P10 Equations: The complete analytical solutions (by integrating the exp functions) + Taylor expansion of the Exp functions may help define non-dimensional numbers.

P14L8-9 Do you mean "storage" in the catchment?

P14L18 Croissant et al. Proposed that t0 is almost independent on Landslide size above some threshold because of river entrenchment and narrowing when they cross landslides. Would this behaviour change the main results? I am surprised that the necessarily shorter tcon of big landslides (because they likely reach the river network) and their large volume do not contribute more to the exported C, although I understand that they are less frequent.

P15L2-3 Nice result, and not so intuitive. That said, the production of C in soil also depends on mean annual runoff (and temperature) but is not taken into account in the model. I guess that this would reinforce this finding as wetter climate would generate more OC in soils. Does the dependence on mean rainfall rate mean that arid landscapes are more likely carbon sources? If this is the case, at global scale, the net budget of organic C may depend on the relative areas of arid and humid landscapes.

P18L5 pb of citation format.

P23 the Fan et al. (2019) reference is doubled.

Best wishes, Sébastien Carretier

---

## Referee Comment (RC2) · Aaron Bufe (Referee) · 21 Dec 2020

Review of Croissant et al., *Pulsed carbon export from mountains by earthquake-triggered landslides explored in a reduced-complexity model* submitted to Earth surface dynamics

**Summary**

I enjoyed reviewing this manuscript and found it not only interesting, but also very carefully organized and well-written. The fate of OC is clearly an important topic, and this contribution provides a new model framework that can be adapted to ask many questions about the fate of OC mobilization by landslides in mountain regions.

Below, I lay out a few points about the model setup and assumptions that left me confused and that the authors may be able to address before publication. Beyond these points, I do not have any major comments or suggestions on this well-crafted piece.

**Sediment transport and fluvial OC export in the model**

As far as I understand, the authors assume that OC is transported as part of the fine sediment fraction. In turn, fine sediment transport is parameterized as a function of the average sediment transport capacity, $\bar{Q}_s$. As per equation (8), did I understand correctly, that for a single landslide body, the transport of sediment and, thus, the removal of $M_{OC}$ just scales directly with this transport capacity of the river at the point that the river passes the deposit? I would have intuitively thought that, in many cases, the river, where it passes the deposit, is already transporting near, or at, capacity – thus, the additional sediment that it can erode when it passes the landslide may not corresponds to the full transport capacity $\bar{Q}_s$. In other words, wouldn't you have to route sediment through the channel network, to determine the excess transport capacity at each of the landslides? The difference between capacity and excess capacity could be quite large? I am probably misunderstanding something, but even if I am, clarifying the assumption in the text may help some readers that have the same thought.

Further, I struggled to follow the jump from the individual equations of $M_{oc,t}$ for fluvial transport and oxidation, to equations 12-14. Here, it could help to first define $M_{ox,t}$: $M_{ox,t} = k_{ox}M_{oc,t}$ (I think). Then you can explicitly state in the text that you combine this equation with eq. 10 and integrate to obtain eq. 12. The same applies to giving an explicit definition of $M_{riv,t}$ before the integration and a statement of which equations you combine and integrate to obtain equations 13 – 14.

Finally, I got confused about the parameterization of fluvial transport in the model. In particular the link between the differential equation (8) and the linear scaling with $\overline{Q_s}$ in equations (11, 13-14) remains unclear to me. As far as I understand, you first give a differential equation for the loss of $M_{oc}$ by river export in equation (8). The solution to this equation, similar to eq (10) for oxidation, is not given (not sure why?), but should be $M_{oc,t} = M_{oc,0} e^{-\frac{\overline{Q_s}}{M_{ls}} t}$. Then, a characteristic time is introduced. This time looks similar to the e-folding time in the solution above, but it is now expressed (1) in terms of volume rather than mass and (2) in reference to only the fine material, not the entire landslide material ($M_{ls}$). Moreover, this timescale is not an e-folding time, but assumed to be the time to remove the entire landslide volume – leading to a linear scaling of sediment export with $\overline{Q_s}$. This jump was not clear to me. What is $M_{riv,t}$? Analogous to $M_{ox,t}$, from equation (8), I would have thought it should be $M_{riv,t} = \frac{\overline{Q_s}}{M_{ls}} M_{oc,t} = M_{oc,0} \frac{\overline{Q_s}}{M_{ls}} e^{-\frac{\overline{Q_s}}{M_{ls}} t}$. Similarly, I would have thought that the combined loss of OC to both oxidation and fluvial transport would just be the sum of eq (8) and eq (9) with the solution: $M_{oc,t} = M_{oc,0} e^{-\left(\frac{\overline{Q_s}}{M_{ls}} + k_{ox}\right) t}$. However, this is not what equations (11, 13-14) show.

So, in summary, it would be very helpful if you could specify why riverine OC export is first introduced via a differential equation with an exponential solution, and then later modeled as an inverse function of $t_0$ or a linear function of $\overline{Q_s}$.

I am sorry, if this is horribly confused, but hopefully these comments can give some ideas on how to clarify the model setup.

**Assumptions in the model**

In P 18, L29 – p19, L21, you discuss the limits to the parameterization of the connection velocity. You also mention the assumption that once the toe of the landslide deposit is connected to the channel, everything is connected. Given the volumes of landslide sediments stored on hillslopes in the field, this assumption strike me as one of the more significant ones. An additional aspect here, that is sort of touched on in the paragraph, may be that, after some time, fine sediment may become limited by the removal of the coarse fraction – either because the coarse sediment shields finer sediment underneath, or because the coarse

sediment at the toe of the landslides stabilizes the remaining material on the hillslope. If that was true, then OC transport may be sensitive also to the transport of coarse material and to the role of extreme events that can mobilize that coarse material. It could be possible to implement it in the model via a different version of equation (8) – such implementation may be beyond the scope of the paper, but discussing this limit and the potential to overestimate (?) OC export could be useful.

Is it worth mentioning the assumption that OC is distributed equally on all slopes (e.g. on P6)? I can imagine that very steep slopes have less OC than gentler slopes – whereas landslides are biased toward steeper slopes. Thus, may landslides be biased toward slopes that have OC stocks below the landscape-average?

On a similar note, by modelling landslides to only occur during earthquakes, the landscape has more time to rebuild the carbon stock between earthquake events. Rainfall-triggered landslides in the Southern Alps are common. Moreover, reactivation of previous slopes (which are necessary, as you say, to deliver all of the sediment to the channel and allow the connection of the entire sediment mass to the channel) may disturb the buildup of soils and biomass on the slopes. Thus, OC stock buildup may be itself a function of the erosion of material from the hillslope. Perhaps it could be valuable to add a note on how reactivation or interseismic landsliding affects the estimates of the model.

**Line & Figure comments**

P2, L29: I believe there should be a hyphen between 'sediment' and 'transfer'?

P3, L1: Why is this relevant only for OC at the surface?

P5, L23: I suggest to explicitly state that it is sediment in the *entire* deposit that is assumed to be connected.

P6, L4: Typo; landslide 'scar' without 's'.

P6, L13 – 15. You could consider adding a reference for this statement.

P10, L22: Space missing

P11, L11: 'soil' without 's'

P16, L19: Is 'sort' supposed to be 'sought'? Otherwise, I do not understand the sentence.

P31, L5: 'is shown'

Fig 5: I wasn't sure whether the histograms referred to the blue or the red line, or to both?

Fig. 9: The relevant parameter is runoff variability which scales with 1/k, so I suggest to replot the figure with 1/k on the x-axis – if this change is not made, I suggest to at least change the label 'runoff variability'. Otherwise, the reader might read high values as high runoff variability.

I hope that the comments are helpful and remain with best wishes to the authors and editor,

Sincerely,

Aaron Bufe

---

## Author Comment (AC1) · 20 Apr 2021

The reply to the reviewers is in the attached supplement.

Please also note the supplement to this comment:
https://esurf.copernicus.org/preprints/esurf-2020-95/esurf-2020-95-AC1-supplement.pdf

---

## Author Response (AR1)

**Reply to reviews: "Pulsed carbon export from mountains by earthquake-triggered landslides explored in a reduced-complexity model" for *Earth Surface Dynamics Discussions**

We thank Sébastien Carretier and Aaron Bufe for their positive appraisal of our manuscript and for their timely comments and suggestions. We reply herein, and have used their reviews to improve the manuscript.

Thomas Croissant and Bob Hilton on behalf of the authors,

**REVIEWER 1 :**

This manuscript is very well written and easy to read. It constitutes a real added value in the understanding of the CO2 budget of mountains with respect to organic carbon, taking into account the spatial and temporal variability of the competition between oxi-dation and physical export of OC at the scale of a catchment (landslides + rivers) and over a seismic cycle. The figures are self-explanatory. The limits of the model are clearly indicated and the conclusions are supported by the results. I think this contribution is close to being in good shape for publication.

Thank you. We have made some important revisions to the paper as a result of your comments and those of Reviewer 2, and reply to your questions and comments herein.

In addition to the specific points listed below, I suggest considering the possibility of presenting part of the results in an a dimensionalised form. For example, according to the formalism of the model, it seems that an important parameter is the dimensionless product to * kox. I therefore wonder whether the curves in Figures 2 and 3 in particular would not collapse by grouping the curves by value of t0 * kox and (t0+tcon) * kox.

We can see the appeal of presenting part of these results (Figure 2 and 3) in a dimensionless form. However, having done proposed calculations from the simulations, we find that they do not help to clarify the main model outputs and findings. In the case of an initially connected landslide (i.e. tcon = 0), all the curves with the same ratio of (t0*kox) do indeed collapse.

However, when we do the same operation but group the curves by ((t0+tcon)*kox), they do not collapse. This is because these simulations include equations that describe both the connected landslides, and disconnected landslides. For example, for a constant value of (t0+tcon) and a constant kox, t0 and tcon can vary as long as their sum is equal. This would involve different lengths of disconnected phases and therefore an asymmetry in the duration of the processes that act on the disconnected and connected phase. As such, we have decided not to include such a transformation in the revised manuscript.

Concerning to, the time of evacuation by rivers, Thomas Croissant has shown in another paper (Croissant et al., 2017, Nat Geo) that the export time of connected landslides no longer significantly depends on the size of the landslide above a threshold, thanks to the lateral dynamics of the rivers incising these landslides. In this contribution, Thomas Croissant et al. do not seem to consider this limitation. This saturation process may reinforce their conclusion of an effective export of OC and may deserve comment.

We answer this question in the 'specific comment' section below. In short, it relates to whether we consider suspended load (here in this study, and its linked fine particulate organic carbon), or bedload sediment export (Croissant et al., 2017).

The study of a seismic cycle seems to consider only one magnitude of earthquakes (7.9) occurring at regular intervals. What would be the contribution of other possible earthquakes to the C balance?

**In the case of the Western Southern Alps of New Zealand, the Alpine Fault exhibits a pattern of repeated large earthquakes that rupture along the fault – resulting in large magnitude earthquakes with a recurrence interval of ~263 years (as discussed in the manuscript with reference to the considerable prior work on this topic). In between these large events, paleoseismic assessments have shown that the Alpine Fault does not produced large earthquakes (i.e. Mw > 5) that would produced a significant sediment delivery to the river network.**

**Because we use the catchment morphology, the prevailing climate of the Western Southern Alps and landslide distribution metrics of this study area, we have decided to focus our case on the example of the measured recurrence interval and estimated earthquake magnitude. One could use our approach to explore what happens if earthquake magnitude-frequency distributions were different. This would be very valuable, however given the existing ground we cover in this paper, we decided that this was out of scope of our paper.**

**That said, we have considered the potential "background" landsliding that reflect rainfall-triggered landslides that are common in this area (Hovius et al., 1997). The 'background' landsliding model is based on the empirical data of Hovius et al, (1997), and our results show that the erosion flux of OC by these landslides is several orders of magnitude lower than the signal produced by large earthquakes in the first decade following the seismic event.**

**Specific comments**

P8L20 Define better what is a "pool" (landslide? Soil layer within a landslide?)

**We have clarified the term « pool ».**

P8L25 Not clear to me how the multi-pool approach used here relates to this observation that kox should depend on age.

**We have clarified this – it is essentially that the 1st pool contains the youngest organic matter, while the 2nd and 3rd pools contain organic matter that has resided in the system for longer periods. We have altered the text here to better explain this.**

P8L27 (and P7L9 and P9L8) Residence time, turn over time etc: These "times" have a precise meaning in the Reservoir Theory underlying this work, and do not necessarily match. If the distribution of OC ages is heavy-tailed, the concept of Residence Time may be not sufficient to characterise the oxidation processor any process relating to the ageing of C (e.g. Mudd and Yoo, JGRES, 2010,https://doi.org/10.1029/2009JF001591). Could you comment on this?

**We were careful in the submitted version not to use the term "residence time" in terms of the organic matter phase, as this means different things to different fields, and how it is calculated. For instance, the $1/k_{ox}$ is a model output that relates to a timescale. However, measured radiocarbon activity (age) does not measure the same thing (Trumbore, 2000), nor do soil incubation experiments (Tate et al., 1995).**

**As the reviewer notes, soil organic matter ages form a heavy tail distribution – this is partly what the multi-pool model is seeking to capture (by having a "slow pool" with lower $k_{ox}$). Or indeed, the continuum model approach, as discussed in Arndt et al., (2013).**

**We have decided not to link to the Mudd and Yoo study here (and others of this type), because this introduces a further concept of the residence time of mineral phases, which is likely to further confuse the matter. Instead, we have removed the specific note P9L8 on "turnover" – and clarified the text here in terms of the $k_{ox}$ metric, and its link to timescale.**

P10 Equations: The complete analytical solutions (by integrating the exp functions) +Taylor expansion of the Exp functions may help define non-dimensional numbers.

**Please refer to the answer above related to the adimentionalization of the equations.**

P14L8-9 Do you mean "storage" in the catchment?

**Yes, we have clarified this in the paragraph.**

P14L18 Croissant et al. proposed that t0 is almost independent on landslide size above some threshold because of river entrenchment and narrowing when they cross landslides. Would this behaviour change the main results? I am surprised that the necessarily shorter tcon of big landslides (because they likely reach the river network) and their large volume do not contribute more to the exported C, although I understand that they are less frequent.

**It is true that accounting for the river entrenchment of the landslide deposits would mean that above a certain landslide size, the relationship between t0 and landslide volume should be lost. However, the case of Croissant et al, (2017) focused on bedload transport capacity. Here we are studying the evacuation of the fine material transported as suspended load, that we infer are located in the surface and subsurface of the landslide deposit and which may be transported independently of the morpho-dynamic changes invoked by Croissant et al., (2017). As such, given the lack of an equivalent theoretical framework for suspended load transport, we have simplified the sediment entrainment and export dynamics.**

P15L2-3 Nice result, and not so intuitive. That said, the production of C in soil also depends on mean annual runoff (and temperature) but is not taken into account in the model. I guess that this would reinforce this finding as wetter climate would generate more OC in soils. Does the dependence on mean rainfall rate mean that arid landscapes are more likely carbon sources? If this is the case, at global scale, the net budget of organic C may depend on the relative areas of arid and humid landscapes.

**Regarding the result here ("the results show more sensitivity to the mean annual runoff than to runoff variability"), we go on to explain why this is the case in terms of the parameterisation of the model. Indeed, it may emerge that runoff variability becomes more important in a threshold-based transport model, as we discuss.**

**In terms of the comment on how "a wetter climate would generate more OC in soils", and how this feeds into our approach. We have added a paragraph to the end of this section which comments on how the climatic setting will also influence the vegetation type and soil development (and soil organic carbon stocks).**

**We also mention this in Section 7.2 when considering the wider applicability of the findings. However, we chose not to explore this in our western Southern Alps example, but it is clearly of interest to this broader question. Further model developments could attempt to capture this dynamic and build it into experiments.**

P18L5 pb of citation format.

**This has been corrected.**

P23 the Fan et al. (2019) reference is doubled.

**We have some issues with reference management software, but will ensure that this is corrected.**

Best wishes,

Sébastien Carretier
* * *
**REVIEWER 2 :**
**Summary**

I enjoyed reviewing this manuscript and found it not only interesting, but also very carefully organized and well-written. The fate of OC is clearly an important topic, and this contribution provides a new model framework that can be adapted to ask many questions about the fate of OC mobilization by landslides in mountain regions.
Below, I lay out a few points about the model setup and assumptions that left me confused and that the authors may be able to address before publication. Beyond these points, I do not have any major comments or suggestions on this well-crafted piece.

**Thank you for these comments and for your thoughtful review.**

**Sediment transport and fluvial OC export in the model**

As far as I understand, the authors assume that OC is transported as part of the fine sediment fraction. In turn, fine sediment transport is parameterized as a function of the average sediment transport capacity, $\overline{Qs}$. As per equation (8), did I understand correctly, that for a single landslide body, the transport of sediment and, thus, the removal of MOC just scales directly with this transport capacity of the river at the point that the river passes the deposit?

**Yes, this is the case. We make sure this is clear in the revision.**

I would have intuitively thought that, in many cases, the river, where it passes the deposit, is already transporting near, or at, capacity – thus, the additional sediment that it can erode when it passes the landslide may not corresponds to the full transport capacity $\overline{Qs}$. In other words, wouldnt you have to route sediment through the channel network, to determine the excess transport capacity at each of the landslides? The difference between capacity and excess capacity could be quite large? I am probably misunderstanding something, but even if I am, clarifying the assumption in the text may help some readers that have the same thought.

**This might be true for bedload transport but not in the case of suspended load, for which the proportion of deposition in the active, steep mountain channel is likely to be minimal.**

**Indeed, in a another paper (currently in preparation), we tested this hypohesis by running landslide evacuation simulations at the catchment scale using a 2D morphodynamic model Eros (see Davy et al, 2017, JGR and Croisant et al 2017, Nature Geoscience for examples of the abilities of the model). In the model Eros, the sediment is routed throughout the fluvial network and is therefore sensitive to variation of transport capacity along the river network. Comparing the eventual fluxes of suspended sediment predicted by Quakos and by Eros at the catchment outlet, we have seen that they are comparable in amplitude and dynamic.**

Further, I struggled to follow the jump from the individual equations of $M_{oc,t}$ for fluvial transport and oxidation, to equations 12-14. Here, it could help to first define $M_{ox,t}$: $M_{ox,t} = k_{ox}M_{oc,t}$ (I think). Then you can explicitly state in the text that you combine this equation with eq. 10 and integrate to obtain eq. 12. The same applies to giving an explicit definition of $M_{riv,t}$ before the integration and a statement of which equations you combine and integrate to obtain equations 13 – 14.

Finally, I got confused about the parameterization of fluvial transport in the model. In particular the link between the differential equation (8) and the linear scaling with $\overline{Q_s}$ in equations (11, 13-14) remains unclear to me. As far as I understand, you first give a differential equation for the loss of $M_{oc}$ by river export in equation (8). The solution to this equation, similar to eq (10) for oxidation, is not given (not sure why?), but should be $M_{oc,t} = M_{oc,0}e^{-\frac{\overline{Q_s}}{M_{ls}}t}$. Then, a characteristic time is introduced. This time looks similar to the e-folding time in the solution above, but it is now expressed (1) in terms of volume rather than mass and (2) in reference to only the fine material, not the entire landslide material ($M_{ls}$). Moreover, this timescale is not an e-folding time, but assumed to be the time to remove the entire landslide volume – leading to a linear scaling of sediment export with $\overline{Q_s}$. This jump was not clear to me. What is $M_{riv,t}$? Analogous to $M_{ox,t}$, from equation (8), I would have thought it should be $M_{riv,t} = \frac{\overline{Q_s}}{M_{ls}}M_{oc,t} = M_{oc,0}\frac{\overline{Q_s}}{M_{ls}}e^{-\frac{\overline{Q_s}}{M_{ls}}t}$. Similarly, I would have thought that the combined loss of OC to both oxidation and fluvial transport would just be the sum of eq (8) and eq (9) with the solution: $M_{oc,t} = M_{oc,0}e^{-\left(\frac{\overline{Q_s}}{M_{ls}}+k_{ox}\right)t}$. However, this is not what equations (11, 13-14) show.

So, in summary, it would be very helpful if you could specify why riverine OC export is first introduced via a differential equation with an exponential solution, and then later modeled as an inverse function of $t0$ or a linear function of $Qs$.

I am sorry, if this is horribly confused, but hopefully these comments can give some ideas on how to clarify the model setup.

**Thanks for this careful tracking through the defining equations. We agree that the presentation and explanation of the equations was not as clear as it could have been. We have rewritten the Sections 2.3 and 2.5 in order to clarify them. We also propose to add a list of defined parameters as an Appendix at the end of the revised paper.**

**[Note – apologies for any formatting issues in this reply – we have copied images of the pdf text to try and avoid this as much as possible.]**

**Assumptions in the model**

In P18, L29 –p19, L21, you discuss the limits to the parameterization of the connection velocity. You also mention the assumption that once the toe of the landslide deposit is connected to the channel, everything is connected. Given the volumes of landslide sediments stored on hillslopes in the field, this assumption strike me as one of the more significant ones. An additional aspect here, that is sort of touched on in the paragraph,maybe that, after some time, fine sediment may become limited by the removal of the coarse fraction –either because the coarse sediment shields finer sediment underneath, or because the coarse sediment at the toe of the landslides stabilizes the remaining material on the hillslope. If that was true, then OC transport may be sensitive also to the transport of coarse material and to the rôle of extreme events that can mobilize that coarse material. It could be possible to implement it in the model via a different version of equation (8) –such implementation

may be beyond the scope of the paper, but discussing this limit and the potential to overestimate (?) OC export could be useful.

**We agree that the processes described here are relevant, but they would require extensive developments on the model and are therefore out of the scope of this paper. However we have added a paragraph in the 'limitations' section of the discussion to include these reflections.**

Is it worth mentioning the assumption that OC is distributed equally on all slopes (e.g. on P6)? I can imagine that very steep slopes have less OC than gentler slopes –whereas landslides are biased toward steeper slopes. Thus, may landslides be biased toward slopes that have OC stocks below the landscape-average?

**Indeed, we have conducted soil sampling and made soil organic carbon (SOC) concentration measurements across a set of sites in the West Coast of New Zealand (a part of which has been published in Wang et al, 2020, Science Advances, and other parts can be found in Harvey, 2019, Landslides and organic carbon erosion: Reassessing the role of landslides as transient carbon stores in the western Southern Alps, New Zealand. Masters thesis, Durham University. http://etheses.dur.ac.uk/13287/). These data suggest that in the West Coast, here is no clear correlation between the organic carbon-rich SOC stocks in the surface ~10 to 20 cm, and any of the classical topographic indexes that might control SOC stock (e.g. elevation , slope, …). Thus in this case, a constant value for OC stock is justified. We have examined the text here and summarised these findings and made a link to the Harvey, 2019, thesis.**

On a similar note, by modelling landslides to only occur during earthquakes, the landscape has more time to rebuild the carbon stock between earthquake events. Rainfall-triggered landslides in the Southern Alps are common. Moreover, reactivation of previous slopes (which are necessary, as you say, to deliver all of the sediment to the channel and allow the connection of the entire sediment mass to the channel) may disturb the buildup of soils and biomass on the slopes. Thus, OC stock buildup may be itself a function of the erosion of material from the hillslope. Perhaps it could be valuable to add a note on how reactivation or interseismic landsliding affects the estimates of the model.

**Yes, this is a valid point, and an important one. We have added a paragraph in the discussion section 5. to discuss the reactivation of landslide scars.**

**Line & Figure comments**

**We thank the reviewer for their careful reading of the manuscript. We have accounted for the proposed corrections in the revised manuscript.**

P2, L29: I believe there should be a hyphen between 'sediment' and 'transfer'?
P3, L1: Why is this relevant only for OC at the surface?
P5, L23: I suggest to explicitly state that it is sediment in the entire deposit that is assumed to be connected.
P6, L4: Typo; landslide 'scar'without 's'.
P6, L13 –15.You could consider adding a reference for this statement.
P10, L22: Space missing
P11, L11: 'soil'without 's'
P16, L19: Is 'sort'supposed to be 'sought'? Otherwise, I do not understand the sentence.
P31, L5: 'is shown'
Fig 5: I wasn't sure whether the histograms referred to the blue or the red line, or to both?
Fig. 9: The relevant parameter is runoff variability which scales with 1/k, so I suggest to replot the figure with 1/k on the x-axis – if this change is not made, I suggest to at least change the label 'runoff variability'. Otherwise, the reader might read high values as high runoff variability.

**All above comments fixed and/or clarified.**

I hope that the comments are helpful and remain with best wishes to the authors and editor,

Sincerely,

Aaron Bufe

---

## Referee Report (RR1)

Review of Croissant et al., *Pulsed carbon export from mountains by earthquake-triggered landslides explored in a reduced-complexity model* resubmitted to Earth surface dynamics

I had found the previous version of the manuscript to be very clear, and I like the new additions that were made. In my second read-through, I particularly focused on the derivation of model equations that I commented on during my last review. It was great to see the model equations being re-explained. Unfortunately, I was still confused by the derivation of the quantities and found them hard to track. After some pondering, I believe that I found the reason for my confusion, but I acknowledge that I might still misunderstand something.

I believe the main confusion stems from a lack of clarity in distinguishing between expressions that quantify the mass remaining in a deposit versus the mass that is removed by river erosion or oxidation. There are two layers to this.

- First, Equations 8 – 15 are almost all expressed in terms of mass remaining in the deposit whereas the final equations (17-19) track the mass removed. This made it hard for me to immediately understand the final equations from the preceding paragraphs.
- Second, the definitions are not consistent (to me). Many quantities that seem to track the mass remaining in the deposit are described as mass removed. In addition to having a number of subscripts and jumps between these, I ended up being confused for a while about what was expressed where.

I try to illustrate this confusion below. This is a long comment which is not meant to be patronizing in any way. Rather, I had to write the equations out for myself in order to be able to structure my thoughts. Perhaps it helps to illustrate my confusion. Importantly, the changes needed to address these points seem very minor. In particular, I suggest to come up with a notation scheme (maybe combining subscripts and superscripts) that makes it very clear whether a given variable tracks the mass removed from the deposit versus the mass remaining under either oxidation, river erosion, or both. Further, you could consider at each step to systematically give expressions for both the mass removed and the mass remaining. Finally, when you get to the final equations, it would really help to clearly say which equations you are combining to arrive at these expressions. This would require adding a couple of expressions for the mass removed which are currently missing.

I hope that my comments are clear. Please feel free to contact me directly if you have any questions about them.

Notations

For the purposes of this review, I use the following notation. I don't want to necessarily suggest to use these particular notations in the manuscript, but hopefully they illustrate what I mean by separating the different masses more systematically. I use upper case $M$ to track mass remaining in the deposit, and lower case $m$ to track masses that are being removed (there might be a better solution).

I follow the manuscript and use subscripts to designate the type of mass

- $M_{oc}$, $m_{oc}$ designate a mass of organic carbon
- $M_{ls}$, $m_{ls}$ designate a mass of fine sediment

I use superscripts to designate the method of removal

- $M^{riv}$, $m^{riv}$ is the evolution of a mass under river erosion
- $M^{ox}$, $m^{ox}$ is the evolution of a mass under oxidation
- $M^{riv+ox}$, $m^{riv+ox}$ is the evolution of a mass under both oxidation and river erosion

I follow the manuscript and use qualifiers to the subscript to designate the point in time

- $M_{oc,t}$, $m_{oc,t}$ designate a mass of organic carbon at time.

On a side note, I find the use of $t_0$ to describe the time it takes to erode the entire landslide volume slightly confusing. In all other cases, the subscript 0 refers to some initial point in time (e.g. the initial mass mobilized by landsliding), so intuitively, I'd think of $t_0$ as the time when the landslide occurred – it is a minor point, but you could consider expressing this differently.

Particular equations

P7 L4 and equation 8: Here, equation (8) is described as: "the mass of fine sediment ($M_{ls}$) being exported by river" (note, there is a 'the' or 'a' missing). To me, $M_{ls}$ and equations (8&9) describe not the mass exported by rivers, but the mass remaining in the deposit – or more fully, the mass remaining in the landslide under the condition that the mass is removed by fluvial erosion. It would still be useful to give the mass removed, because you end up using the mass removed to obtain equation (17), I think. I.e.

- Mass of sediment removed at any point in time: $m_{ls,t}^{riv} = \overline{Q_s} = \dfrac{M_{ls,0}}{t_0}$

- Evolution of sediment remaining in the deposit: $\dfrac{dM_{ls}^{riv}}{dt} = -m_{ls,t}^{riv} = -\overline{Q_s}$ which then solves to yield the mass remaining in deposit at any point in time $M_{ls,t}^{riv} = \overline{Q_s}(t_0 - t)$.

P8 L18 and equation 10: Here you could define the mass more clearly: $M_{oc}$ is the mass of OC remaining in the landslide as far as I understand.

P9 L18-22 and equation 11: Here, you solve equation (10), but all of a sudden, the subscript of the quantity that is solved changes from $M_{oc}$ in equation (10) to $M_{ox}$ in equation (11). Also, the definition of the quantity changes from the mass remaining in the landslide to the mass that is oxidized in the landslide. This jump is not clear and it is not clear why the same expression would describe the mass removed and the mass remaining. Maybe it is just a typo, because you have it expressed correctly in equation (16). In any case, I think it would help a lot to be consistent in your definitions of mass removed and mass remaining and then give both quantities. I.e.

1. Mass that is oxidized at any point in time: $m_{oc,t}^{ox} = k_{ox} M_{oc,t}$

2. Evolution of OC mass in the landslide under oxidation only: $\frac{dM_{oc}^{ox}}{dt} = -m_{oc,t}^{ox} = -k_{ox} M_{oc}$, which when solved gives the mass that is remaining in the landslide under oxidation only: $M_{oc,t}^{ox} = M_{oc,0} e^{-k_{ox}t}$.

3. Therefore, the mass that is oxidized at any point in time: $m_{oc,t}^{ox} = k_{ox} M_{oc,0} e^{-k_{ox}t}$. This is different from equation (11) by a factor of $k_{ox}$

P9 L27 – P10 L3 and equations 12 & 13: Again, the way that these equations are parameterized, the quantity $M_{riv,t}$ seems to describe the mass of organic carbon that is remaining in the landslide under the condition that mass is removed by fluvial erosion. However, it is described in the text as "the mass of OC exported by rivers". Again, there seems to be some inconsistency between definition and equation. Moreover, the subscript _riv is not very intuitive for a mass of OC remaining in the deposit. Finally, in equation (17) $M_{riv}$ is actually the mass removed. In other words, the parameter $M_{riv}$ in equations 12/13 and the same parameter in equation 17 describes two different quantities. Finally, I would, again, consider giving both mass removed and mass remaining, because you need the latter to get to equation (17). I think it would be:

- Mass of OC removed at any point in time: $m_{oc,t}^{riv} = \overline{Q_s} \frac{M_{oc,t}}{M_{ls,t}}$

- Evolution of OC in the deposit $\frac{dM_{oc,t}^{riv}}{dt} = -m_{oc,t}^{riv} = -\overline{Q_s}\frac{M_{oc,t}}{M_{ls,t}}$ which then solves to yield the mass of OC remaining in the deposit at any point in time assuming removal by river erosion: $M_{oc,t}^{riv} = M_{oc,0}\left(1 - \frac{t}{t_0}\right)$

*NOTE that there seems to be a minus sign too many in equation (13), the solution to equation (12)*

Equations (17)-(18). For these two equations, I suggest to consider explicitly building them from the equations and definitions you give before. This would require to give expressions for the masses removed. For example, equation (16) seems to be the integral of $m_{oc,t}^{riv} = \overline{Q_s}\frac{M_{oc,t}}{M_{ls,t}}$ and then substituting $M_{oc,t}^{riv+ox}$ from your equation (15) and $M_{ls,t}^{riv}$ from your equation 9 – and then integrating the equation between $t_{con}$ and $t_0$ with the initial mass that is modulated by the oxidation during the unconnected phase. It took me quite a bit of time to make that connection, probably in part because I was confused about tracking the masses. However, I think it would have helped me to have the derivation explicit.

For both equation (17) and (18), shouldn't the integral be from the connection time rather than time zero? I.e. $\int_{t_{con}}^{t_0} dt$ instead of $\int_0^{t_0} dt$ ?

Finally, I got confused about $M_{land}$ – in particular, why we need another parameter here. $M_{land}$ is defined as the mass of sediment remaining in the deposit. In that sense, what is difference between $M_{land,t}$ and $M_{oc,t}$ as defined in equation (15)? I understand that in equations 17-19, these quantities are now the total quantities integrated across time. However, when integrating up to $t_0$, which, per definition, is the time to remove the entire deposit, shouldn't $M_{land}$ per definition be just zero? I may be missing something.